# The epigenetic pioneer EGR2 initiates DNA demethylation in differentiating monocytes at both stable and transient binding sites

Karina Mendes[1,3], Sandra Schmidhofer[1,4], Julia Minderjahn[1,5], Dagmar Glatz[1,6], Claudia Kiesewetter[1,7], Johanna Raithel[2], Julia Wimmer[1,8], Claudia Gebhard[2] & Michael Rehli [1,2 ✉]

The differentiation of human blood monocytes (MO), the post-mitotic precursors of macrophages (MAC) and dendritic cells (moDC), is accompanied by the active turnover of DNA methylation, but the extent, consequences and mechanisms of DNA methylation changes remain unclear. Here, we profile and compare epigenetic landscapes during IL-4/GM-CSF-driven MO differentiation across the genome and detect several thousand regions that are actively demethylated during culture, both with or without accompanying changes in chromatin accessibility or transcription factor (TF) binding. We further identify TF that are globally associated with DNA demethylation processes. While interferon regulatory factor 4 (IRF4) is found to control hallmark dendritic cell functions with less impact on DNA methylation, early growth response 2 (EGR2) proves essential for MO differentiation as well as DNA methylation turnover at its binding sites. We also show that ERG2 interacts with the 5mC hydroxylase TET2, and its consensus binding sequences show a characteristic DNA methylation footprint at demethylated sites with or without detectable protein binding. Our findings reveal an essential role for EGR2 as epigenetic pioneer in human MO and suggest that active DNA demethylation can be initiated by the TET2-recruiting TF both at stable and transient binding sites.

[1] Department of Internal Medicine III, University Hospital Regensburg, 93053 Regensburg, Germany. [2] Regensburg Centre for Interventional Immunology, University Hospital Regensburg, 93053 Regensburg, Germany. [3] Present address: Universidade Católica Portuguesa, Center for Interdisciplinary Research in Health (CIIS), Institute of Health Sciences (ICS), Viseu, Portugal. [4] Present address: AstraZeneca, 22880 Wedel, Deutschland. [5] Present address: Sandoz GmbH, 6336 Langkampfen, Austria. [6] Present address: Chromatin Structure and Cellular Senescence Research Unit, Maisonneuve-Rosemont Hospital Research Centre, Montréal, QC H1T 2M4, Canada. [7] Present address: Labor Kneissler, 93133 Burglengenfeld, Germany. [8] Present address: Deutsches Patent- und Markenamt, 80331 München, Germany. ✉email: michael.rehli@ukr.de

DNA methylation turnover is an essential epigenetic process in normal hematopoiesis, and its dysregulation favors leukemogenesis[1,2]. In hematopoietic cell types, TET2 is the most relevant hydroxylase catalyzing the iterative oxidation of 5-methylcytosine (5mC) that initiates its demethylation[3–5]. It is required for active DNA demethylation processes in proliferating hematopoietic progenitor cells[6–8] as well as post-mitotic human blood monocytes (MO)[9,10].

Despite recent advances in understanding the physiological relevance of TET enzymes, the processes controlling TET2 recruitment to individual target loci in hematopoietic cells remain unclear. TET2 lacks the CXXC domain that enables direct CpG-binding of TET1 and TET3 to DNA, suggesting that it depends on transcription factors (TF) or their co-factors for locus-specific recruitment[11]. Concordantly, several TF were reported to interact with TET2, including Wilms Tumor 1 (WT1), which was found to recruit TET2 to its target genes in acute myeloid leukemia (AML) cells[12,13], or early B-cell factor 1 (EBF1), which was reported to interact with TET2 in IDH-mutant cancers[14]. PU.1 was also reported to interact with TET2 during human MO to osteoclast differentiation[15]. In the context of reprogramming of murine B cells into induced pluripotent cells, CEBPA, KLF4, and TFCP2L1 were shown to recruit TET2 to specific DNA sites, leading to enhancer demethylation and activation during reprogramming[7].

To fully understand how DNA methylation landscapes are established during normal hematopoiesis or malignant transformation, we need to understand the mechanisms that control site-specific de novo DNA methylation and demethylation processes. An ideal model to study the latter is the differentiation of human peripheral blood MO. Upon isolation and ex vivo culture, human MO differentiate into various MO-derived cell types depending on cytokines, growth factors, or other environmental cues[16–19] in the absence of cell division[9], which precludes passive DNA demethylation mechanisms. Previous reports documented active DNA demethylation events in GM-CSF/IL-4-driven MO differentiation[9,10,20]. In this model, MO differentiate into so-called MO-derived dendritic cells (moDC), originally described by Sallusto and Lanzavecchia[16], which were shown to share features of ascites-derived inflammatory moDC[21]. The marked morphological, transcriptional, and epigenetic changes during ex vivo culture were shown to coincide with site-specific TET2-dependent active DNA demethylation, but it was unclear how TET2 is recruited to demethylated sites.

This study investigates the mechanisms and TF involved in active DNA demethylation during the GM-CSF/IL-4-driven differentiation of human peripheral blood MO in vitro. Genome-wide analyses of DNA methylation turnover as well as functional studies highlight the importance of two demethylation-associated TF (interferon regulatory factor 4 (IRF4) and early growth response 2 (EGR2)) in IL4/GM-CSF-driven MO differentiation and reveal a particular role for EGR2 in active DNA demethylation during this process. Interestingly, the site-specific TET2 recruitment by EGR2 was not restricted to sites that were detectably bound by this factor. Our findings suggest that transient interactions of epigenetic pioneers like EGR2 (that do neither result in chromatin remodeling as detected by ATAC, nor in ChIP-detectable binding) are sufficient to induce the active demethylation process around their binding sites.

## Results

**Genome-wide DNA methylation changes during MO differentiation.** The cytokine-driven, in vitro differentiation of primary post-mitotic human blood MO has previously been established as an excellent model to study active DNA demethylation[9,10]. To better understand the mechanisms driving the DNA methylation turnover during IL-4/GM-CSF-mediated differentiation of MO into immature dendritic cells (moDC) as well as the consequences of the epigenetic remodeling, we first obtained a global view of DNA methylation changes to correlate them with transcription, chromatin accessibility, and TF binding (Fig. 1a). The comparison of published whole-genome bisulfite sequencing (WGBS) data sets for MO ($n = 4$)[22] and moDC ($n = 6$)[23] revealed extensive losses and only limited gains in DNA methylation (Fig. 1b), confirming previous observations on smaller scales[9,10,20,24].

In total, we identified 7731 differentially methylated regions (DMR). Of those, 7610 regions reproducibly lost DNA methylation during moDC differentiation (DMR demethylated in moDC), while only 121 regions gained DNA methylation (DMR methylated in moDC, Fig. 1c). Since MO and moDC data sets were from independent sources, we randomly selected DMR from both groups and determined their DNA methylation status in additional matched donor samples generated in our lab. The changes at DMR methylated in moDC were generally not reproducible (Supplementary Fig. 1a, b), suggesting that they mainly represent technical artifacts, while DMR demethylated in moDC were consistent across data sources and donors. We therefore focused our further analyses on DMR demethylated in moDC, which were enriched in promoters, exons, and transcription termination sites (TTS) (Fig. 1d).

As previously observed[10], only a fraction of DMR overlapped with genes that were significantly regulated during MO differentiation (Fig. 1e), indicating that active demethylation processes occur independent of transcriptional changes. The distribution of 5-hydroxymethylcytosine (5hmC) across demethylated DMR was globally assessed using a 5hmC capture-seq approach via sequential glycosylation and biotinylation. It followed the expected patterns: lower in MO with strong increase after 18 h of culture, and decreased but broader distribution in moDC (Fig. 1f, g; left panels). Hydroxylation was generally acquired rather than lost during differentiation (Supplementary Fig. 1c). In line with the unidirectional DNA methylation turnover, regions that lost chromatin accessibility maintained their DNA methylation state while those that gained accessibility were demethylated (Supplementary Fig. 1d). We also observed an increase in chromatin accessibility across demethylated DMR at many regions (Fig. 1f, g; right panels). However, half of the DMR were below the ATAC-seq peak threshold (Supplementary Fig. 1e), suggesting that many but not all DMR become accessible during MO differentiation. Previously identified (*DNASE1L3*, *CLEC10A*)[10] and novel example loci (*SLAMF1*, *MAF*) demonstrating local active demethylation are shown in Fig. 1h and Supplementary Fig. 1f. DMR demonstrating differentiation-associated changes in 5hmC distribution as well as changes in open chromatin are highlighted.

**Effect of TET2-depletion on MO differentiation.** We have previously shown that siRNA-mediated knock-down can delay DNA demethylation in differentiating MO[9], but due to toxicities arising from the original transfection procedure we were unable to study the functional consequences of TET2 knock-down. Here, we established a transfection protocol for human MO using backbone-modified siRNAs that appears to have no impact on MO survival and does not activate the cells; allowing to study functional consequences of TET2 loss during differentiation. The knock-down of TET2 was almost complete after 3 days (Supplementary Fig. 2a), but it had no impact on cell survival (Fig. 2a) or cell morphology (Supplementary Fig. 2b) of moDC. DNA demethylation was significantly delayed in TET2 knock-down cells as confirmed by EpiTYPER DNA methylation analysis

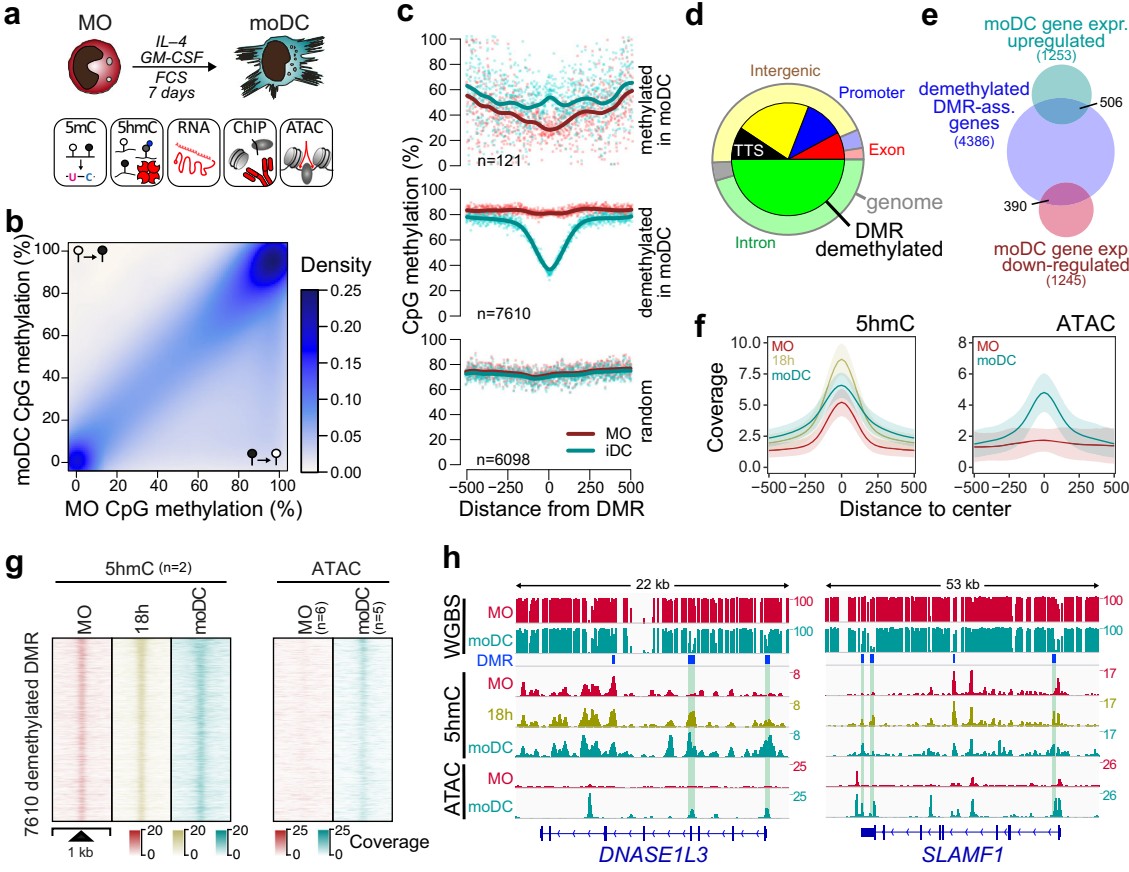

**Fig. 1 DNA demethylation during MO differentiation correlates with dynamic changes in 5hmC and increasing chromatin accessibility. a** Schematic of the experimental setup for in vitro MO differentiation into moDC and downstream methodologies. **b** Scatterplot showing the density distribution of average DNA methylation ratios (WGBS) in MO (n = 4) and moDC (n = 6). **c** Genomic distance distribution of averaged DNA methylation ratios (at single CpG resolution) centered on differentially methylated regions (DMR), methylated or demethylated in moDC and random regions. Lines represent spline curves across individual data points. Sample types are indicated by coloring. **d** Pie chart illustrating the genomic location distribution of DMR demethylated in moDC (inner circle) relative to the entire genome. **e** Venn diagram depicting the overlap between genes associated with DMR demethylated in moDC and genes up- or downregulated during GM-CSF/IL4-driven MO differentiation. **f** Genomic distance distribution of 5hmC enrichment and ATAC coverage centered on DMR demethylated in moDC. Shaded bands represent the 95% confidence intervals. **g** Distribution of the 5hmC and ATAC-seq signals across DMR demethylated in moDC in MO and moDC. **h** IGV genome browser tracks of example regions (*DNASE1L3* and *SLAMF1* loci) including CpG methylation ratios (WGBS), 5hmC and ATAC-seq coverage in MO and moDC, as well as the genomic locations of DMR. Highlighted DMR (green boxing) indicate regions demonstrating increased chromatin accessibility and increased 5hmC deposition during differentiation. (**c**, **d**, **f**, **g**) Source data are provided as a Source Data file.

for several DMRs that are known targets of DNA demethylation (Fig. 2b and Supplementary Fig. 2c). Next, we analyzed both transcriptional changes and changes on the level of chromatin accessibility in TET2 knock-down cells. On the global level, transcriptional programs in TET2 knock-down cells were distinguishable from control cells (Supplementary Fig. 2d), but we only detected few genes that were significantly affected by TET2 knock-down across replicates (downregulated: 37 genes; upregulated: 12 genes; FDR ≤ 0.1, RPKM ≥ 1, heatmap is shown in Fig. 2c). The set of downregulated genes was significantly enriched for genes that are induced during moDC differentiation (Fig. 2d), indicating that the TET2 knock-down selectively affected a small subset of genes induced during MO differentiation. Similar observations were made on the level of chromatin accessibility. Out of 106K accessible regions that were detected across replicates in control and TET2 knock-down cells, only 11 were significantly different across replicates in TET2 knock-down cells. A focus on DMR overlapping ATAC-peaks, however, showed a clear tendency towards losing accessibility upon TET2 knock-down as compared with control regions that are not

changing accessibility during MO differentiation (Fig. 2e). EpiTYPER DNA methylation analysis for several regions loosing accessibility upon TET2 knock-down also confirmed a delay of DNA demethylation at these sites (Supplementary Fig. 2e). Collectively, the data suggest a limited effect of siRNA-mediated TET2-depletion on transcription and chromatin accessibility. RNA-sequencing-derived gene expression profiles of the other two TET family members (Supplementary Fig. 2f) suggest that TET3 might compensate for TET2 activity during MO differentiation.

**TF associated with active demethylation events.** In MO, the DNA methylation at DMR demethylated in moDC ranging from fully methylated to only partially methylated (>50%). To further analyze their properties, we therefore divided DMR into quintiles with decreasing methylation ratios in MO (Fig. 3a). A comparison with corresponding DNA methylation levels in CD34+ hematopoietic progenitor cells[22] (Supplementary Fig. 3a) suggested that DMR demethylated in moDC represent a mixture of

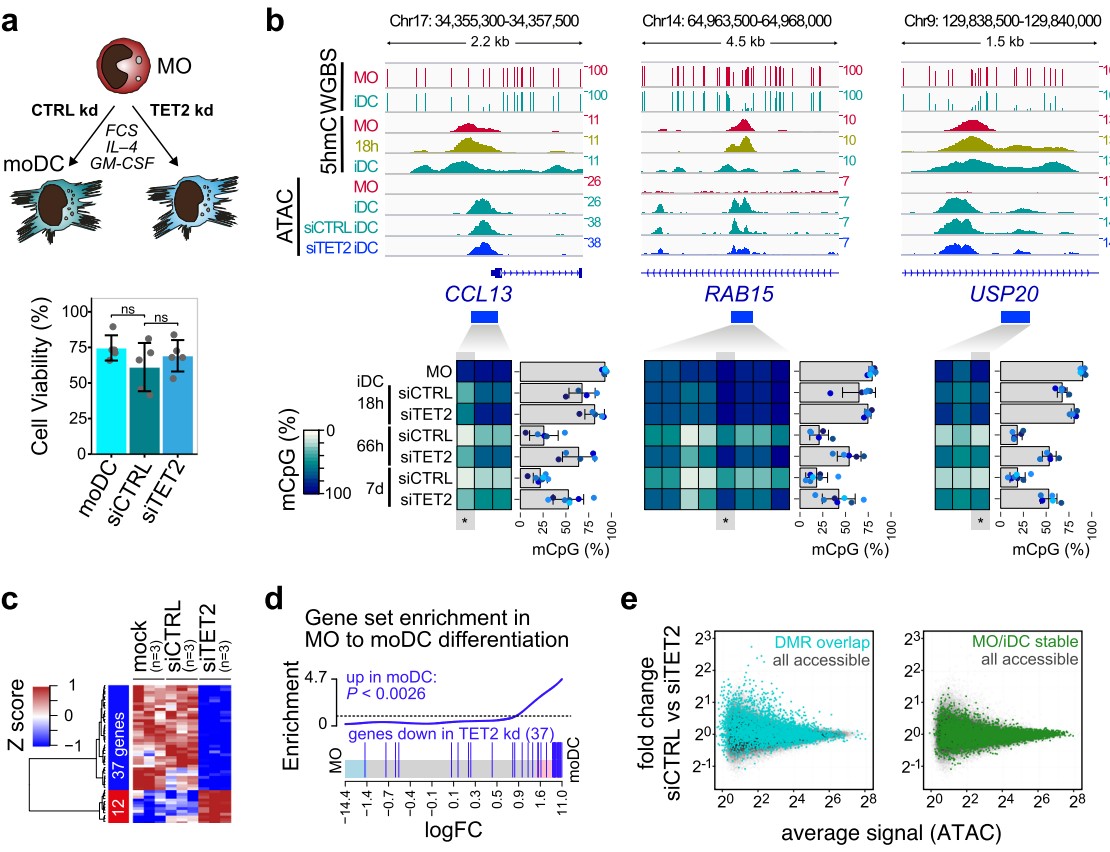

**Fig. 2 Impact of TET2 knock-down (kd) on cell viability, gene expression, chromatin accessibility, and DNA methylation. a** Schematic of the experimental setup and effect of siTET2 treatment on moDC viability relative to the number of cultured monocytes. Bars in the bottom panel represent mean ± SD of $n = 5$ biologically independent experiments. Shown are values for untransfected moDC as well as siRNA-transfected cells (siCTRL, siTET2). Individual data points are shown as gray dots (ns, not significant, paired $t$-test, two-sided). **b** IGV genome browser tracks for the indicated WGBS, 5hmC, and ATAC data sets at regions around example DMR that were previously identified in moDC[10]. Corresponding DNA methylation ratios for the indicated regions are given below. For heatmaps, methylation ratios (representing means of $n = 5$ (18 h and 66 h) or $n = 7$ (MO and 7 d) biologically independent experiments) are indicated by coloring (white: no methylation, dark blue: 100% methylation, gray: not detected) with each column representing a single CpG. For each region the data of a single CpG (highlighted and marked by asterisks) is shown. Bars represent mean ± SD, individual data points are shown as colored dots (each color representing a different donor). **c** Heatmap presenting scaled expression data of differentially expressed genes (FDR $\leq$ 0.1, RPKM $\geq$ 1) between control- (siCTRL) and siTET2-treated cells. **d** Barcode plot demonstrating the enrichment of the set of genes downregulated in siTET2-treated cells in moDC (compared to MO). The enrichment $P$-value of the two-sided rotation gene set test is indicated. **e** MvA plots for ATAC-seq data comparing control (siCTRL) and siTET2 treatment. Coloring indicates ATAC peaks overlapping DMR (top panel) or ATAC peaks that remain stable (invariant) during MO differentiation. (**a**–**e**) Source data are provided as a Source Data file.

de novo demethylated regions and regions that initiated demethylation processes already at the MO stage, or even at earlier progenitor stages. De novo motif searches across DMR revealed five motifs corresponding to known candidate TF, including a PU.1 consensus motif, a composite ETS/IRF motif, as well as AP1, STAT, and EGR motifs (Fig. 3b). This motif signature is similar to the motifs derived from moDC-specific ATAC peaks (Supplementary Fig. 3b). Interestingly, the distribution of each motif varied with the methylation state in MO. Partially methylated regions in MO (progressively demethylated sites, indicated by the light red pies) were more enriched for PU.1 and AP1 motifs, while fully methylated sites in MO (de novo demethylated sites, indicated by the dark red pies) were more highly enriched for STAT and EGR motifs. This is in line with the induction of the latter factors upon culture, while the former are already present in MO. The composite motif (ETS/IRF response element, EIRE), which is likely recognized by PU.1:IRF heterodimers, also showed a higher enrichment at regions partially methylated in MO.

Analysis of the RNA-seq data for MO and moDC suggested that *IFR4* and *EGR2* are transcriptionally activated in moDC

(Supplementary Fig. 3c). Further analyses confirmed their induction on protein level (Supplementary Fig. 3d), qualifying them as candidate factors driving de novo demethylation processes.

To study the actual genomic distribution of our candidate TF we performed ChIP-seq of native PU.1 protein as well as FLAG-tagged versions of PU.1, EGR2, and IRF4 that were introduced via mRNA electroporation to compensate for the lack of suitable antibodies against EGR2 and IRF4. Comparing DMR and TF bound or accessible sites, we identified three groups of DMR (Fig. 3c). Two-thirds of DMR overlapped with one or more TF peaks (blue coloring), while roughly one-sixth were accessible (but not bound by PU.1, IRF4, or EGR2, yellow coloring), and the remaining DMR were neither accessible nor bound (red coloring). The latter ("no peak") group still showed the typical 5hmC gain over time (Fig. 3c) and was clearly demethylated (Fig. 3d, top panel) although to a lesser degree compared to TF-bound DMR (Fig. 3d, bottom panel). Motif searches identified the same signature TF motifs across the three DMR groups (Supplementary Fig. 3e). However, motifs in the "open" and the "no peak" groups showed far fewer co-occurrences compared to the "TF peak" group as shown in the co-association networks presented in

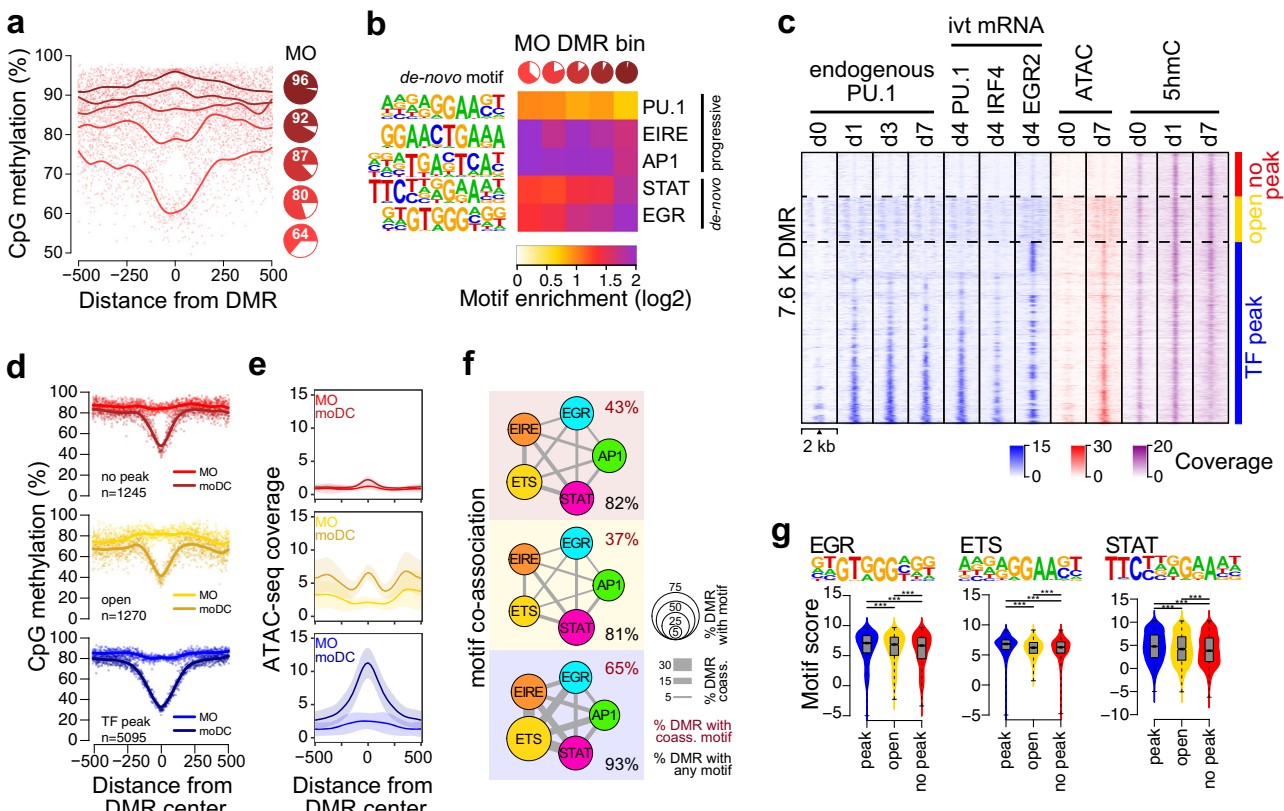

**Fig. 3 TF signatures at DMR. a** Genomic distance distribution of mean methylation ratios in MO centered on DMR demethylated in moDC. Pie charts represent the mean methylation level (in red) of DMR quantiles (1.5K each) sorted by their mean methylation ratio in MO. **b** Heatmap of motif enrichment (compared to background) for the indicated TF contingent on DMR quintiles based on the methylation state of MO (pie charts shown on the top and in (**a**)). **c** Distribution of ChIP-seq (native PU.1 and FLAG-tagged versions of PU.1, IRF4, and EGR2 derived from transfected, in vitro-transcribed (ivt) mRNA), ATAC-seq, and 5hmC signals across the 7610 DMR demethylated in moDC at the indicated time points of moDC differentiation. DMR are divided into three majors groups characterized by detectable TF binding (blue, "TF peak"), the presence of open chromatin without detectable TF binding (yellow, "open"), and the absence of open chromatin and lack of detectable TF binding (red, "no peak"). **d** Genomic distance distribution of averaged DNA methylation ratios in MO and moDC centered on DMR demethylated in moDC in "TF peak" (n = 5058), "open" (n = 1270), and "no peak" (n = 1245) groups. **e** Genomic distance distribution of ATAC-seq coverage in MO and moDC centered on DMR demethylated in moDC in "TF peak", "open", and "no peak" groups. Shaded bands represent the 95% confidence intervals. **f** Motif co-association networks for "TF peak", "open", and "no peak" groups. The size of each node represents the motif enrichment (fraction of peaks) and co-associated TF motifs are indicated by coloring. Edge thickness indicates the frequency of motif co-association. The fraction of DMR with co-associated motifs and the fraction of DMR with any motif are given above and below each network, respectively. **g** Combined bean and box plot showing EGR, ETS, and STAT motif log-odds score distributions for "TF peak", "open", and "no peak" groups. Solid bars of boxes display the interquartile ranges (25–75%) with an intersection as the median; whiskers represent max/min values. Significantly different motif score distributions in pairwise comparisons are indicated (***P < 0.001, Mann–Whitney U test, two-sided). (**a**, **b**, **d**–**g**) Source data are provided as a Source Data file.

Fig. 3f. In addition, motif scores for EGR, AP1, and STAT were generally higher in the "TF peak" group (Fig. 3g), indicating that the "open" and the "no peak" groups are characterized by lower affinity motifs that are also sparser across these DMR. Interestingly, open but unbound regions (yellow coloring) showed a unique ATAC-seq pattern (Fig. 3e, central panel) suggesting that the DMR-associated accessible sites associate with closely neighboring accessible sites (separated by a single nucleosome). Motif searches revealed distinct enrichment patterns across the central and adjacent accessible regions: open chromatin regions overlapping the central demethylated area were enriched for AP1, EGR, and STAT motifs, while adjacent open chromatin regions were primarily enriched for ETS (PU.1) and composite ETS:IRF (EIRE) elements (Supplementary Fig. 3f).

**Effect of TF-depletion on MO differentiation and DNA demethylation.** The DMR-associated motif signatures and TF-binding maps strongly suggested that TF like EGR2 or IRF4 may

be responsible for the recruitment of TET2 to their binding sites to initiate hydroxylation of 5mC. The knock-down of IRF4 in MO efficiently prevented its upregulation during differentiation (Supplementary Fig. 4a), had no impact on cell survival (Fig. 4a), but drastically altered cell morphology. Instead of the typical non-adherent, irregularly shaped appearance of moDC with dendritic projections, IRF4-depleted cells frequently presented a "fried-egg" morphology resembling adherent MO-derived macrophages (MAC) (Supplementary Fig. 4b). In line with morphological changes, IRF4-depletion induced a MAC-like gene expression program, comparable to MO-derived MAC (Fig. 4b). Genes upregulated by IRF4 knock-down were clearly enriched in genes upregulated during the differentiation of MO-derived MAC (Fig. 4c). Genes downregulated by IRF4 knock-down were enriched for gene ontology terms associated with moDC functions (Fig. 4d), including antigen presentation. A comparison of expression profiles for selected marker genes of MAC and moDC is provided in Supplementary Fig. 4c.

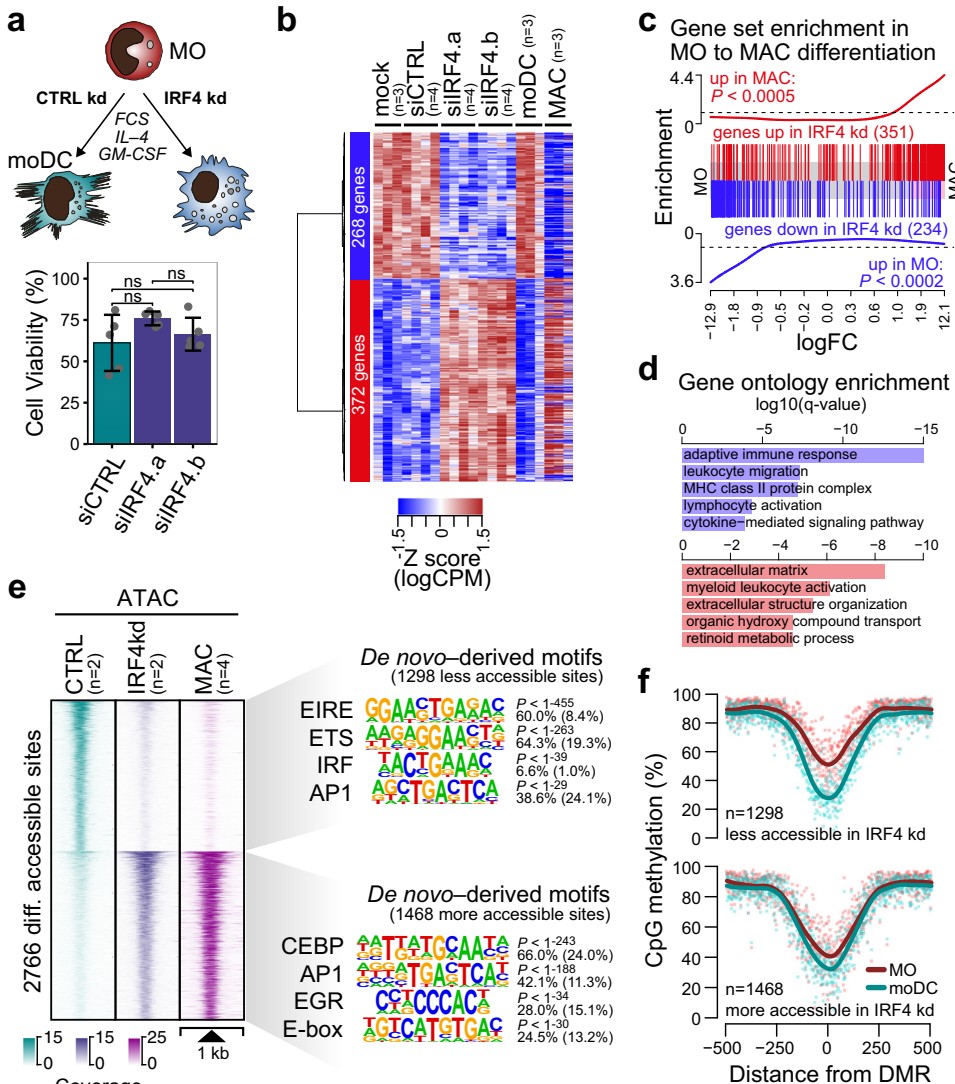

**Fig. 4 Effect of IRF4 knock-down on MO differentiation. a** Schematic of the experimental setup and effect of siIRF4 treatment on moDC viability. Bars in the bottom panel represent mean ± SD of $n = 5$ biologically independent experiments. Individual data points are shown as gray dots (ns, not significant, paired $t$-test, two-sided). **b** Heatmap presenting hierarchically clustered and scaled expression data of differentially expressed genes in control- (mock, siCTRL) versus siIRF4-treated cells (absolute logFC > 1, logCPM and logRPKM > 1, and FDR < 0.05). Each column corresponds to an individual donor. For comparison, expression data of independent cultures of moDC and MAC were included into the heatmap before scaling. **c** Barcode plots showing the enrichment of the indicated gene sets (red: up in siIRF4-treated cells; blue: in siIRF4-treated cells) across the logFC ranked gene list of MO to MAC differentiation. Enrichment $P$-values of two-sided rotation gene set tests are given. **d** Gene Ontology (GO) terms associated with genes upregulated (red bars) or downregulated (blue bars) in siIRF4-treated compared to control-treated cells, as analyzed by Metascape. Bars represent corrected, log-transformed $P$-values ($q$-values) of the GO term enrichment. **e** Distribution of ATAC-seq signals across the differentially accessible sites between control- and siIRF4-treated samples. ATAC-seq signals of MO-derived MAC are plotted across the same regions for comparison. De novo-derived motifs for each cluster are given along with the significance of motif enrichment (hypergeometric test) and the fraction of motifs in peaks (background values are in parenthesis). Top motifs corresponding to known factor families are shown for each cluster. **f** Genomic distance distribution of averaged DNA methylation ratios in MO and moDC centered on differentially accessible sites (as introduced in (**d**)). (**a–f**) Source data are provided as a Source Data file.

Accessible sites in chromatin that were gained after IRF4-depletion were strongly enriched for TF motifs, which were previously defined as being part of a macrophage-enhancer signature, including C/EBP, AP1, EGR2, and E-box motifs[25] (Fig. 4e, bottom motifs). Sites that were lost upon IRF4-depletion were enriched for ETS:IRF composite motifs (EIRE) as well as AP1, PU.1, and IRF motifs (Fig. 4e, top motifs). These data strongly suggested that in human MO, the induction of IRF4 is essential for transcriptional programs controlling key functions of moDC. This phenotype resembles similar observations in the murine system[26].

Focusing on DNA methylation changes during MO differentiation, sites that lose accessibility in IRF4-depleted cells were targets of the demethylation machinery during normal MO differentiation (Fig. 4f). However, many of these sites already showed a significant loss of methylation at the MO stage, suggesting that most sites affected by the IRF4 knock-down already initiated DNA demethylation during earlier stages of MO development.

Compared to IRF4, the effect of EGR2 knock-down—which was most evident at 66 h and 7 days of culture (Supplementary Fig. 5a)—was even more pronounced and clearly affected the

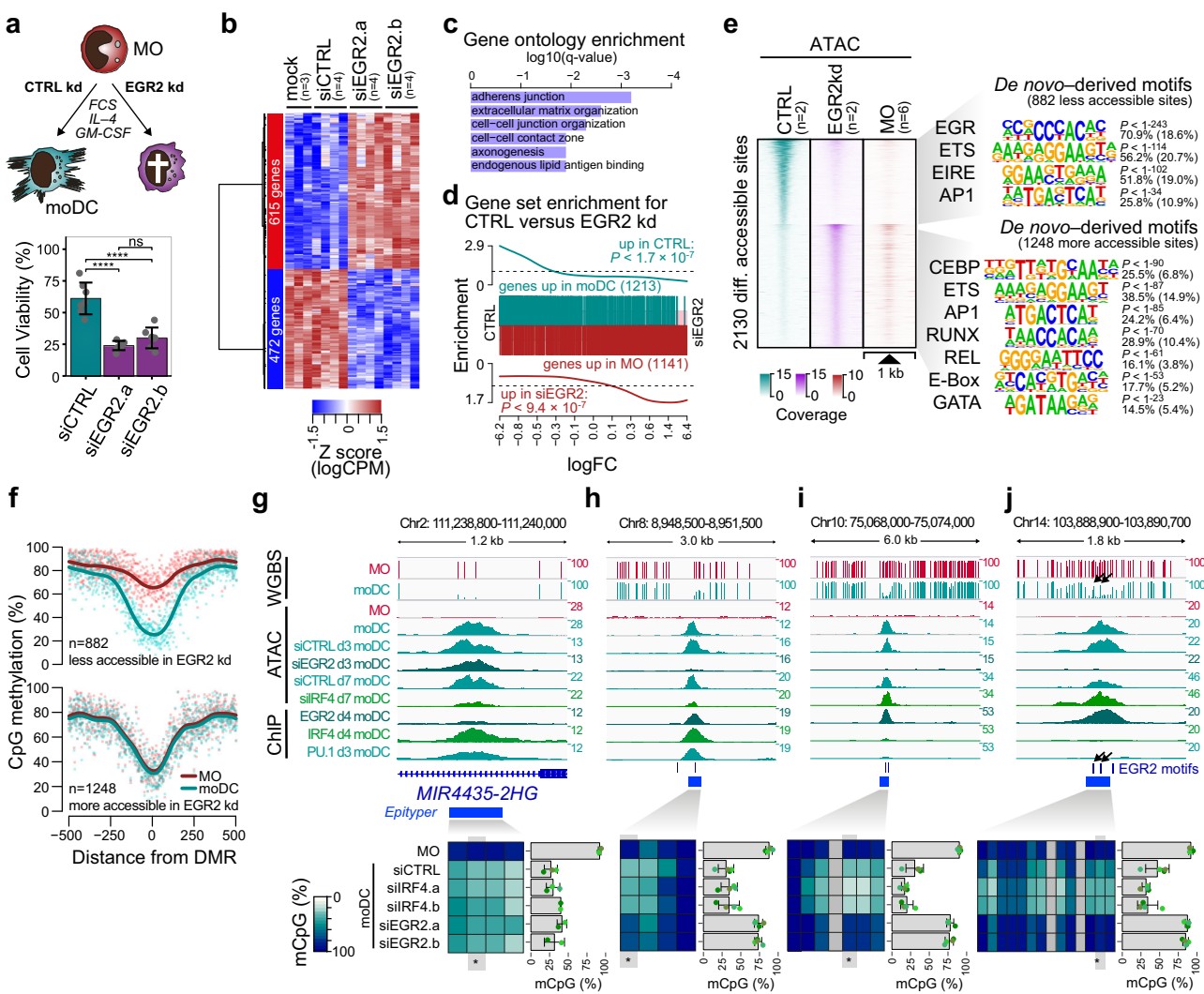

**Fig. 5 Effect of EGR2 knock-down on MO differentiation, survival, and DNA demethylation. a** Schematic of the experimental setup and effect of siEGR2 treatment on moDC viability. Bars in the bottom panel represent mean ± SD of $n = 7$ biologically independent experiments. Individual data points are shown as gray dots (ns, not significant; ****$P < 0.0001$, paired $t$-test, two-sided). **b** Heatmap presenting hierarchically clustered and scaled expression data of differentially expressed genes in control-treated (mock, siCTRL) versus siEGR2-treated cells (absolute logFC > 1, logCPM and logRPKM > 1, and a FDR < 0.05). Each column corresponds to an individual donor and each row represents a single differential expressed gene. **c** Gene Ontology (GO) terms associated with genes downregulated in siEGR2-treated compared to control-treated cells, as analyzed by Metascape. Bars represent corrected, log-transformed $P$-values ($q$-values) of the GO term enrichment. **d** Barcode plots presenting gene set enrichment analysis results across the logFC ranked gene list of siEGR2-treated compared to control-treated cells. Tested gene sets include genes up- or downregulated during moDC differentiation. Enrichment $P$-values of two-sided rotation gene set tests are given. **e** Distribution of ATAC-seq signals across the differentially accessible sites between siCTRL- and siEGR2-treated samples. ATAC-seq signals of MO are plotted across the same regions for comparison. De novo–derived motifs for each cluster are given along with the significance of motif enrichment (hypergeometric test) and the fraction of motifs in peaks (background values are in parenthesis). Top motifs corresponding to known factor families are shown for each cluster. **f** Genomic distance distribution of averaged DNA methylation ratios in MO and moDC centered on differentially accessible sites (as introduced in (**e**)). **g–j** IGV genome browser tracks for the indicated WGBS, ATAC, and ChIP data sets at example regions around binding sites of EGR2, IRF4, or PU.1. Corresponding DNA methylation ratios for the indicated regions are given below. For heatmaps, methylation ratios (representing means of $n = 4$ biologically independent experiments) are indicated by coloring (white: no methylation, dark blue: 100% methylation, gray: not detected) with each column representing a single CpG. For each region, the data of a single CpG (highlighted and marked by asterisks) are shown. Bars represent mean ± SD, individual data points are shown as colored dots (each color representing a different donor). Black arrows (in (**j**)) highlight 5mC-spikes overlapping EGR2 motifs. (**a–c**, **e–j**) Source data are provided as a Source Data file.

survival of differentiating MO (Fig. 5a). While moDC-like morphology was preserved in some of the remaining cells, we also observed cells with few or no typical dendrites as well as apoptotic cells (Supplementary Fig. 5b). Transcriptional programs in EGR2 knock-down cells were clearly distinct from IRF4-deficient cells (Supplementary Fig. 5c), which may be partially explained by an enrichment of contaminating lymphoid cells that were unaffected (and relatively increased) by the EGR2 knock-

down. Hence, genes upregulated in EGR2 knock-down cells were significantly enriched in lymphoid gene sets (Supplementary Fig. 5d) including lymphoid TF (Supplementary Fig. 5e), which limits the interpretability of this gene set. Genes downregulated by EGR2 knock-down (which are less affected by contaminating lymphoid cells) included functional categories related to moDC biology (Fig. 5c). For example, the upregulation of four CD1 molecules that are required for lipid presentation of moDC was

strongly impaired in EGR2 knock-down cells (Supplementary Fig. 5e). In addition, gene set enrichment analysis indicated that genes downregulated by EGR2 knock-down are strongly enriched for genes that are normally upregulated during moDC differentiation while genes upregulated by EGR2 knock-down are enriched in genes normally downregulated during moDC differentiation (Fig. 5d).

To further characterize the impact of EGR2 deficiency, we studied differences in chromatin accessibility between control and EGR2 knock-down cells. Here, monocyte-derived cells were analyzed after 3 days of culture to avoid high background due to apoptotic cells. The comparison of changes in accessibility patterns at highly induced genes (Supplementary Fig. 5f) in control and EGR2 knock-down cells suggest that IL4/GM-CSF-related TF programs at least partially proceed in the absence of EGR2. Many regions, including promoters and putative enhancers, developed moDC-characteristic accessibility patterns, even in the absence of EGR2. Regions failing to gain accessibility in the absence of EGR2 maintained their inaccessible state of MO and showed a strong overlap with a EGR2 consensus motif (Fig. 5e). In addition, they were also enriched in co-associated motifs for PU.1, ETS:IRF (EIRE) or AP-1 (Fig. 5e, top motifs). In line with the gene expression data, the motif signature in regions that failed to lose accessibility during MO differentiation included TF families that are important for myeloid and lymphoid cell biology (Fig. 5e, bottom motifs).

Regions that failed to gain accessibility in EGR2-depleted cells were also markedly demethylated during normal MO differentiation, with many regions being targets of de novo demethylation (Fig. 5f). The much stronger impact of EGR2 compared to IRF4 was also observed when focusing on accessible sites that were lost upon knock-down of either factor (Figs. 4e and 5e) and showed evidence for binding of the respective factor (Supplementary Fig. 5g).

To compare the actual effect on DNA methylation we selected a number of DMR that were bound by EGR2 or IRF4 and determined their DNA methylation status in knock-down and control cells (Fig. 5g–j and Supplementary Fig. 5h–k). Interestingly, we observed regions normally bound by IRF4 that failed to gain accessibility during MO differentiation upon IRF4 knock-down, but were still demethylated (Fig. 5g, h); suggesting that chromatin remodeling and DNA demethylation at these sites are uncoupled and dependent on different TF. In contrast, regions normally bound by EGR2 remained methylated and failed to become accessible during MO differentiation upon EGR2 knock-down (Fig. 5h–j). Collectively, these data implicate EGR2 in the process of active DNA demethylation.

**Initiation of DNA demethylation at transient and stable EGR2-binding sites.** We noted that CpGs within the EGR motif could be protected from demethylation, which resulted in a 5mC-spike within otherwise actively demethylated regions (Fig. 5j). The systematic analysis of all EGR motifs containing DMR demonstrated that CpGs within the EGR core consensus generally remain methylated (Fig. 6a, left panel), while CpGs in another consensus motif (AP1) were demethylated (Fig. 6a, right panel). Notably, the same type of EGR methylation footprint was detected in all three groups of DMR, including the group of TF-bound DMR (TF peak: blue), the group that is accessible (open: yellow) as well as the group of DMR lacking evidence for chromatin accessibility or TF binding (no peak: red). Individual examples for protected CpGs in EGR2 motifs are given in Fig. 6b as well as Supplementary Fig. 6, including additional DNA methylation data across EGR2 consensus sequences.

The nearly complete resistance to demethylation within the EGR2 consensus motifs suggests that the demethylation machinery is recruited to and demethylates surrounding CpGs, but fails to access CpGs overlapping the EGR2 binding sequence. Because demethylation of CpGs around EGR2-binding sites proceeds during MO differentiation while the 5mC-spike at EGR2-binding sites remains methylated, the protecting factor (EGR2) must be present at all times when the demethylation machinery is present and active. The observed pattern is also consistent with a critical role of EGR2 for recruiting the demethylation machinery.

To test whether EGR2 or its known co-factor NAB2 could interact with TET2, we performed co-immunoprecipitations (CoIPs) in a myeloid cell line (THP-1) as well as in moDC. Due to the lack of suitable antibodies against EGR2 or NAB2, CoIPs were performed using cells transfected with mRNAs encoding the FLAG-tagged proteins. TET2 precipitated EGR2 as well as NAB2, and conversely, both EGR2 and NAB2 specifically precipitated TET2, suggesting that the proteins are part of a complex in vivo (Fig. 6c).

The fact that we observe the same methylation footprints at EGR2-bound consensus sequences and at EGR2 consensus sequences with no detectable EGR2 binding nor transposase-accessible chromatin suggests that EGR2 binds the latter sites transiently, and that a transient interaction is sufficient to recruit TET2 (Fig. 6d) which induces the demethylation of surrounding CpGs. Hence, active DNA demethylation can be observed at both stable and transient TF-binding sites (as exemplified by EGR2) that are not necessarily detected via chromatin accessibility or ChIP.

**DNA methylation-spikes are lineage-specific and enriched for consensus motifs of epigenetic modifier TF.** The observed DNA methylation footprints at EGR2-binding sites suggest that at certain loci, methylated CpGs can be protected from TET-mediated demethylation by the TET-recruiting TF. To test whether this is a general phenomenon, we first identified and compared 5mC-spikes (which may indicate demethylation protection) in MO and moDC (Fig. 7a). The regions around moDC-spikes were less methylated compared to MO (Fig. 7b) and slightly enriched for promoter and TTS regions (Fig. 7c). Like DMR, 5mC-spikes in moDC show limited overlap with Tn5-accessible chromatin (Fig. 7d). As expected, moDC-specific 5mC-spikes were enriched for the EGR consensus motif (Fig. 7e), while common and MO-specific 5mC-spikes were highly enriched for CEBP and KLF consensus motifs. Interestingly, both CEBP and KLF family factors were recently identified as epigenetic pioneers that recruit TET proteins to induce DNA demethylation[7], suggesting that DNA methylation-spikes overlap with TF-binding sites. Extending the analysis to additional cell types (including neutrophils, B cells, T cells, NK cells, and hepatocytes) based on published WGBS data sets, we observe the strong cell-type-specific enrichment of particular TF motifs, including CEBP in MO and neutrophils, EGR2 in moDC, EBF in B cells, and ATF/Jun in neutrophils (Fig. 7f). A significant overlap of ChIP-seq peaks for CEBPα or β in human liver, MO, or MAC (Fig. 7g), suggests that 5mC-spikes can be protected from DNA methylation by the bound TF. As described for EGR2, we also observed typical 5mC-spikes at CEBP motifs with no detectable CEBP binding, suggesting that transient CEBPα or β binding may also be sufficient to induce active DNA demethylation at some of its binding sites (Fig. 7h).

**Discussion**

Here we present a comprehensive analysis of DNA methylation changes during the post-mitotic, IL4/GM-CSF-driven differentiation of human blood MO. We determined thousands of

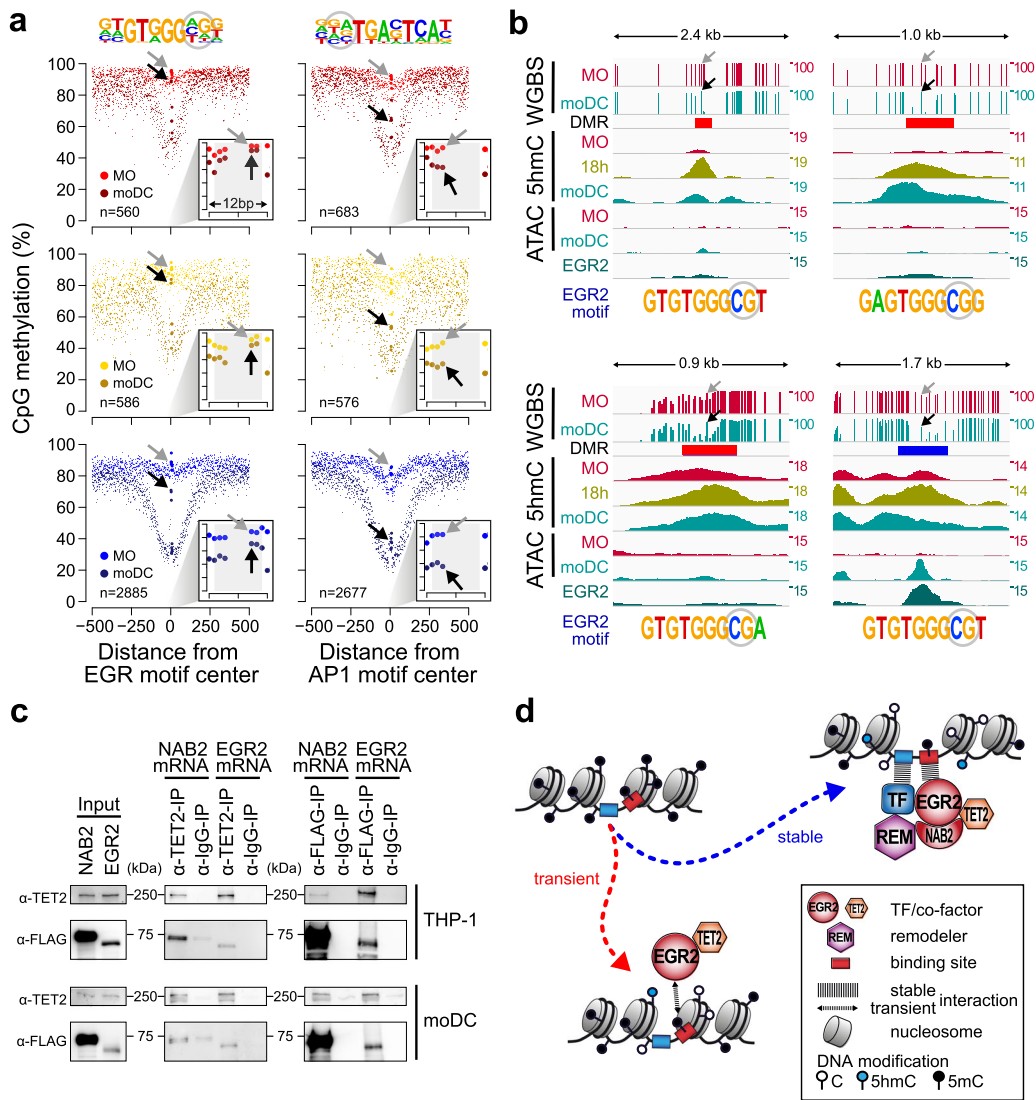

**Fig. 6 Methylation footprint and TET2 interaction of EGR2. a** Genomic distance distribution of averaged DNA methylation ratios in MO and moDC across "TF peak", "open", and "no peak" groups (in blue, yellow, and red, respectively, as defined in Fig. 3c) of DMR demethylated in moDC. DMR were centered on the consensus EGR (left panel) or AP1 motifs (right panel), which are shown on top of each panel (CpG positions that are highlighted by arrows in gray (MO) and black (moDC) in the lower panels are circled). Each panel shows the distribution of DNA methylation ratios across a 1-kb window (large graph) as well as a 12-bp window focusing on the motif-covered CpGs only (smaller inlay graph in the bottom right corner). The number of motifs containing DMR is given in the lower left corner of each panel. **b** IGV genome browser tracks of example loci showing single CpG methylation ratios (WGBS), 5hmC and ATAC-seq coverage in MO and moDC, and EGR2 ChIP-seq coverage in moDC. Genomic locations of DMR are shown (coloring according to the classification into DMR groups "no peak" or "TF peak" in red or blue, respectively). EGR motif sequences are given below each track and arrows in the top tracks mark the positions of the circled CpG. **c** CoIPs of EGR2 or NAB2 (expressed as FLAG-tagged proteins via mRNA electroporation) and TET2 in THP-1 and moDC. Results are representative of two independent experiments. **d** Model of EGR2-induced DNA demethylation at transient and stable binding sites in moDC. (**a**, **c**) Source data are provided as a Source Data file.

actively demethylated sites, defined their characteristics, and identified key regulators of differentiation-associated epigenetic processes. We find that EGR2 is essential for IL4/GM-CSF-driven MO differentiation and also acts as an epigenetic pioneer TF recruiting the DNA demethylation machinery to its binding sites even in the absence of ATAC- or ChIP-detectable chromatin remodeling or TF binding, respectively. We also provide evidence that the latter feature of EGR2 is shared by other epigenetic pioneer TFs that leave a methylation footprint in the absence of detectable binding.

The comparison of epigenetic, chromatin, and TF landscapes allowed us to define the key signature of TF families associated with active DNA demethylation events in MO differentiation, including PU.1, AP1, STAT-family TF, as well as IRF4 and EGR2.

Here we focused on the latter two TF, representing an induced TF associated with early (progressive) demethylation events (IRF4) and a TF associated with late (de novo) demethylation events (EGR2). IRF4 had previously been implicated in MO differentiation towards moDC[26–30]. Confirming earlier mouse work[26], we show that IRF4-depleted human MO acquire a macrophage-like morphology upon culture in IL4/GM-CSF and lose moDC characteristic gene expression signatures, without affecting cell survival. IRF4-depletion clearly impacted chromatin accessibility, but active DNA demethylation proceeded regardless of changes in accessibility, suggesting that remodeling and demethylation are uncoupled; and that the latter may not require IRF4. The majority of IRF4-binding sites in moDC are composite ETS:IRF (EIRE) elements that require PU.1 for IRF4 to bind. PU.1, however, is

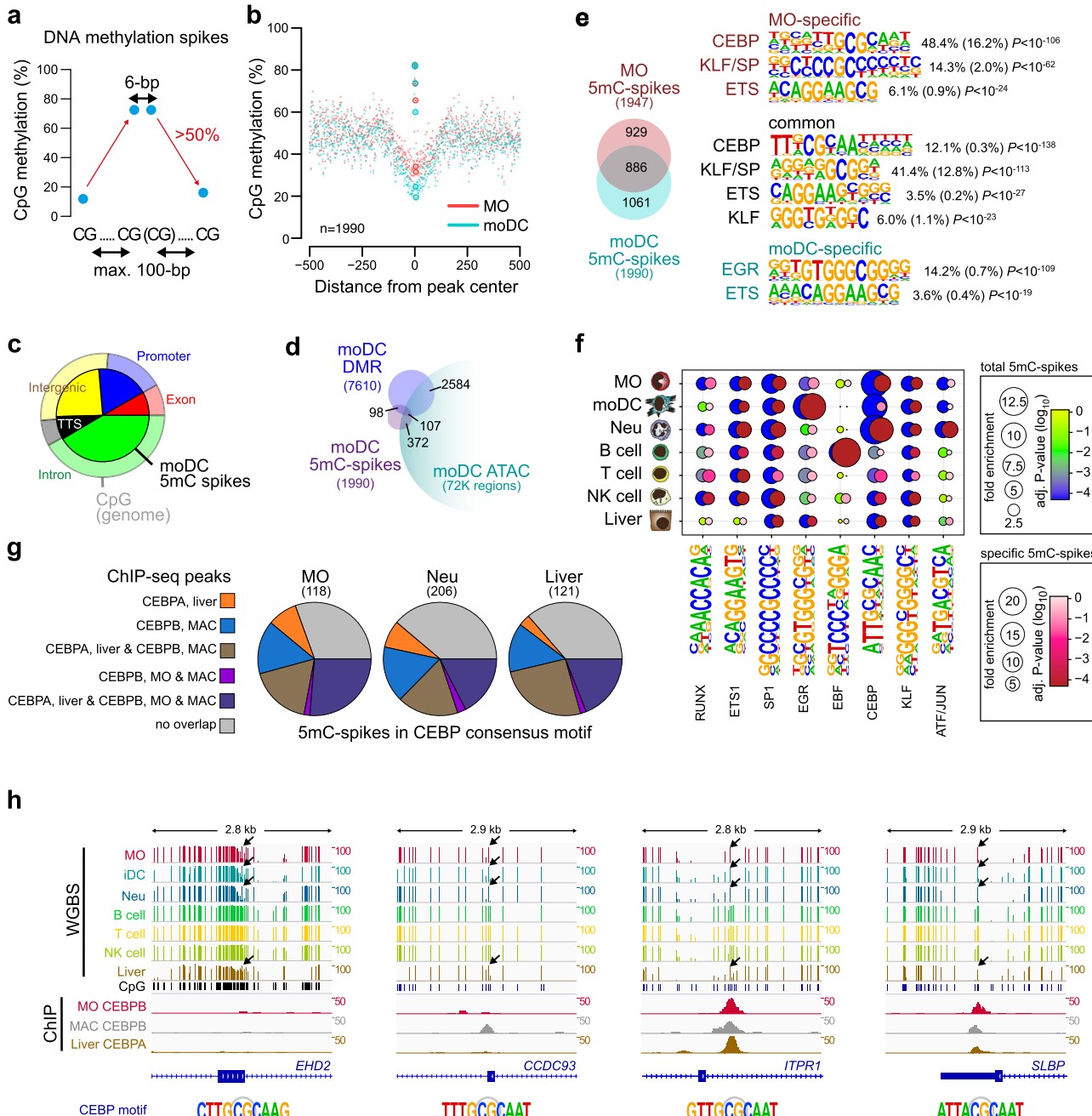

**Fig. 7 DNA methylation-spikes are cell-type-specific and enriched for TF motifs. a** Strategy for identifying small (maximum 6-bp wide) spikes of DNA methylation. **b** Genomic distance distribution of averaged DNA methylation ratios in MO and moDC across moDC 5mC-spikes. Values of central CpGs are indicated by larger dots and the number of 5mC-spikes is given in the lower left corner. **c** Pie chart illustrating the genomic location distribution of moDC 5mC-spikes (inner circle) relative to CpGs across the entire genome. **d** Venn diagram depicting the overlap between 5mC-spikes, DMR, and accessible regions (ATAC peaks) in moDC. **e** Venn diagram indicating the number of common and cell-type-specific 5mC-spikes between MO and moDC. De novo–derived motifs for each 5mC-spike set are given along with the significance of motif enrichment (hypergeometric test) and the fraction of motifs in peaks (background values are in parenthesis). Top motifs corresponding to known factor families are shown for each set. **f** Balloon plot depicting the motif enrichment of selected motifs across all 5mC-spikes identified in each cell type (blue-green-yellow coloring according to corrected enrichment *P*-value) or cell-type-specific 5mC-spikes (red-pink-white coloring according to corrected enrichment *P*-value). The balloon size represents the fold-enrichment and the coloring indicates the corrected *P*-value (hypergeometric test, Benjamini-Hochberg multiple testing correction) of the motif occurrence in 5mC-spikes. **g** Pie charts illustrating the overlap of 5mC-spikes with the indicated CEBP factor ChIP-seq peaks in MO, MAC, and liver. **h** IGV genome browser tracks for the indicated WGBS and ChIP data sets at example regions around 5mC-spikes overlapping the CEBP consensus sequence. (**b**, **c**, **f**, **g**) Source data are provided as a Source Data file.

known to interact with and recruit TET2 to its binding sites, likely regardless of the IRF4 interaction. Hence, while IRF4 is governing moDC-specific transcriptional programs, we found little evidence for its requirement in active DNA demethylation.

EGR2 has previously been implicated in the differentiation of human MO-derived MAC[25], but a role in IL4/GM-CSF-driven MO differentiation into moDC has not been demonstrated. EGR2 is known as an important transcriptional regulator of immune cells that can act either as repressor or as activator of gene expression[31–33]. EGR2 associates with the NAB2 corepressor[34] and early work in a murine model for macrophage differentiation suggested that EGR2 represses neutrophil-specific genes during monocytic maturation[35,36]. However, NAB2 was also identified as an essential co-factor for the recruitment of the mediator subunit INST13 to EGR1/2-bound enhancer elements in human monocyte/MAC[37] and more recent studies in mice have implicated EGR2 in the regulation of transcriptional networks in alternatively activated MAC[38] downstream of STAT6[39]. EGR2 is also required for the upregulation of key factors driving T cell anergy[40], and associated with genes upregulated during in vitro differentiation of human MO into MAC[25]. Hence, its role in hematopoietic cells may depend on cell type and developmental stage.

The presence of its consensus motif in DMR demethylated in moDC as well as its ability to recognize and bind methylated DNA, however, make ERG2 an ideal candidate for an epigenetic pioneer in human MO. In line with this, the knock-down of EGR2 resulted in a marked effect on transcription, chromatin accessibility, and DNA methylation as well as MO survival, suggesting an essential role for EGR2 in IL4/GM-CSF-driven MO differentiation. Genes downregulated upon EGR2 knock-down were primarily enriched for functional categories involving actin cytoskeleton organization, which is known to critically affect moDC functions like T cell adhesion and activation[41–43], phagocytosis[44], or migration behavior[45,46]. Besides its newly uncovered fundamental role for MO biology, EGR2 is also critical for demethylation processes. DMR overlapping EGR2 motifs were mainly late, de novo targets of active DNA demethylation that remained methylated upon EGR2 knock-down. In addition, EGR2 was found to co-immunoprecipitate with TET2, suggesting that EGR2 is required for the recruitment of the demethylation machinery to its binding sites during MO differentiation.

TET2 plays an important role in myelopoiesis, somatic mutations in *TET2* are frequent in myeloid neoplasms[47–50]. TET2 is also the main enzyme responsible for active DNA demethylation processes in MO-derived cells, in particular during in vitro IL4/GM-CSF-mediated MO differentiation into moDC[9]. The siRNA-mediated knock-down of TET2 in MO, in contrast to TF knock-downs, had a very mild effect on chromatin structure and differentiation, which may suggest that TET2 is largely dispensable for the differentiation process of human blood MO. However, a delayed effect of siRNA knock-down on TET2 protein levels and the fact that many demethylated sites initiate demethylation at earlier stages in MO development may attenuate the effect of TET2 knock-down in this system. TET3 may also partially compensate for the loss of TET2 given that it is also expressed during MO differentiation (Supplementary Fig. 2f) and has recently been implicated in regulating the balance of classical and non-classical mouse MO[51]. Notably, our genome-wide comparison of epigenetic, chromatin structure, and TF binding shows that a significant fraction of genomic regions are actively demethylated during MO differentiation without detectable chromatin remodeling or stable TF binding and many more demethylated regions are not associated with altered transcriptional outputs. It is possible that these regions are primed for later usage when appropriate signaling events are triggered to render them functional, but their demethylation may also proceed without

functional consequence. Nevertheless, from a mechanistic point of view, these regions are particularly interesting, because we clearly detect active demethylation processes around TF-binding sites that are not detected by ATAC-seq (as also recently reported for KLF4 in cellular reprogramming[7]), and also not detected by ChIP-seq. The presence of a central, protected 5mC footprint in EGR2 recognition sequences suggests that the epigenetic pioneer EGR2 can shape the DNA methylation landscape via transient interactions and that active DNA demethylation and chromatin accessibility changes are uncoupled at these sites.

Finally, we show that 5mC footprints indicating DNA demethylation protection (identified as 5mC-spikes) are not only observed at EGR2-binding sites. The 5mC-spikes are cell-type-specific and often associated with consensus motifs for known epigenetic pioneer factors like CEBP, KLF of EBF family members. Hence, the ability to induce DNA demethylation via transient protein DNA interaction may not be restricted to EGR2 but represent a more general feature of several epigenetic pioneer factors.

## Methods

**Cell culture**. Collection of blood cells from healthy donors was performed in compliance with the Helsinki Declaration. All donors signed an informed consent. Blood sampling, the leukapheresis procedure, and subsequent purification of peripheral blood MO were approved by the ethical committee of the University of Regensburg (reference number 12-101-0260). Blood MO were separated by leukapheresis of healthy donors followed by density gradient centrifugation over Ficoll/Hypaque and subsequent counter current centrifugal elutriation[52]. For in vitro differentiation of moDC, MO were cultured in RPMI 1640 (Thermo Fisher Scientific, Waltham, USA) supplemented with 280 U/ml GM-CSF (Berlex, Seattle, USA), 20 U/ml IL-4 (Promokine, Heidelberg, Germany)), and 10% fetal calf serum (Sigma-Aldrich, Taufkirchen, Germany). For in vitro differentiation of MO-derived MAC, MO were cultured in RPMI 1640 (Thermo Fisher Scientific, Waltham, USA) supplemented with 2% human AB serum (Bavarian Red Cross, Munich, Germany)[9,10]. Cells were cultured at a density of $1 \times 10^6$/ml and harvested at different time points (18 h, 3d, 4d, and 7d) during differentiation. The human monocytic leukemia cell line THP-1 (DSMZ: #ACC 16) was cultured at a density of $0.5 \times 10^6$/ml as previously described[53]. Cell viability was monitored using trypan blue staining after harvesting.

**In vitro transcription of mRNA**. Synthetic DNA templates (gBlocks) for FLAG-tagged, wild-type EGR2, NAB2, IRF4, and PU.1 were ordered from IDT (Integrated DNA Technologies). Sequences are provided in Supplementary Table 1. Constructs were cloned into pEF6 (Invitrogen) and in vitro-transcribed (ivt) mRNA was generated as previously described[53].

**Transient transfections**. Transient transfections were performed using ivt mRNA (PU.1, IRF4, EGR2, NAB2) or chemically modified siRNAs (Axolabs) against TET2 (si1212), EGR2 (si1132 and si2665), IRF4 (si2384 and si2931), and luciferase (control siRNA). Transfections of mRNAs were performed at 3–4 d after culture and harvested 5–6 h later. All siRNA transfections were performed immediately after MO isolation prior to differentiation and cells were harvested at different time points during culture. Sequences of siRNAs are listed in Supplementary Table 2. Before electroporation with synthetic mRNAs or siRNAs, cells were washed once with phenol red-free RPMI 1640 (Gibco) and once with phenol red-free Opti-MEM I (Gibco) at room temperature. After centrifugation, cells were gently resuspended in phenol red-free Opti-MEM at a final concentration of $3 \times 10^6$ cells in a final volume of 200 μl. For electroporation, in a 4 mm cuvette 200 μl of resuspended cells were combined with 3 μg siRNA or with 5 μg (3×FLAG-EGR2; 3×FLAG-IRF4) and 3 μg (3×FLAG-PU.1) of ivt mRNAs (for ChIP-seq experiments). Cells were electroporated using a Gene Pulser Xcell (BioRad) with a rectangular pulse of 400 V and 5 ms duration. Transfected cells were placed in cell flasks containing pre-warmed culture medium ($1 \times 10^6$/ml) and cultured at 37 °C, 5% $CO_2$, 95% humidity.

**Mass spectrometry analysis of bisulfite-converted DNA**. Genomic DNA was isolated using the DNeasy Blood and Tissue Kit (Qiagen) according to the manufacturer's instructions. DNA was bisulfite converted using the EZ DNA Methylation Kit (Zymo Research) with an alternative conversion protocol (Agena). The bisulfite-converted DNA was amplified by PCR using region-specific primers (designed using the MethPrimer web tool (https://www.urogene.org/methprimer/)[54] and purchased from Sigma-Aldrich). Primer sequences and genomic locations are provided in Supplementary Table 3 (reverse primers were tagged at the 5′ end with the T7 promoter sequence and a 10 mer overhang was added to forward primers to balance for melting temperature differences). PCR was followed by treatment with shrimp

alkaline phosphatase, in vitro transcription, RNA-specific cleavage (MassCLEAVE, Sequenom), and MALDI-TOF MS (MassARRAY Compact MALDI-TOF, Sequenom) as previously described[9,10].

**5hmC capture-seq**. To measure 5-hydroxymethylcytosine (5hmC) in MO and moDC across the genome, we used the Hydroxymethyl Collector kit (Active Motif) according to the manufacturer's instructions. Captured DNA fragments were subjected to library preparation as follows. DNA was end-repaired, A-tailed, adapter ligated (NEXTflex DNA Barcodes), and purified using magnetic beads (Agencourt AMPure XP). Libraries were PCR-amplified (4 cycles), bead-purified, and size-selected on the Caliper LabChip XT (PerkinElmer) using the LabChip XT DNA 300 assay kit (PerkinElmer) according to the manufacturer's instructions. After size selection, DNA was subjected to another 12 cycles of PCR amplification and purified using magnetic beads (Agencourt AMPure XP). The quality of dsDNA libraries was analyzed using the High Sensitivity D1000 ScreenTape Kit (Agilent) and libraries were sequenced on a HiSeq1000 sequencer (Illumina). Sequencing libraries are listed in Supplementary Table 4.

**Co-immunoprecipitation**. CoIPs were performed using anti-TET2 (MABE462, Merck), anti-FLAG M2 (F3165, Sigma), or anti-IgG (sc-2025, Santa Cruz) antibodies (5 μg/CoIP) covalently coupled to Dynabeads M-270 Epoxy beads (1 mg/CoIP, Thermo Fisher Scientific). For each CoIP, $1.5 \times 10^7$ cells were transfected with ivt mRNAs for 3×FLAG-EGR2 (26 μg) or 3×FLAG-NAB2 (29 μg) and incubated at 37 °C, 5% CO$_2$, and 95% humidity. After 3 h, transfected cells were harvested and washed with ice-cold PBS (including PMSF). For cell lysis, 1 ml of Dyna-IP buffer (20 mM Tris-HCl (pH 7.5), 150 mM NaCl, 1% Triton X-100, 2 mM EDTA, 1× protease inhibitor cocktail) was added to each sample, followed by incubation on ice for 30 min with gentle shaking. Then, cell lysates were sonified three times, 10 s each followed by 30 s rest on ice, using a Branson Sonifier 250 (constant duty cycle, output control 2) and insoluble material was removed by centrifugation. The protein concentration of lysates was determined using the Qubit Protein Assay Kit (Thermo Fisher Scientific) and 10 μg of each lysate was kept as input control. Lysates were incubated with anti-IgG- (control), anti-FLAG- or anti-TET2-coupled Dynabeads for 2.5 h at room temperature on a rotating wheel. Protein complexes were washed three times each with PBS including 0.02% Tween-20 and ultrapure water, and eluted from the beads with 50 μl of 2× SDS sample buffer without 2-mercaptoethanol followed by incubation at 95 °C for 10 min. Supernatants were collected on a magnetic rack and 2.5 μl of 2-mercaptoethanol was added to each sample. CoIP and input samples were separated by SDS-PAGE on 8% polyacrylamide gel and the immunoblotting was performed as described below. Blots were stained with the following antibodies: anti-TET2 (A304-247A, Bethyl, 1:1000) and anti-FLAG M2 (F3165, Sigma, 1:2000).

**Immunoblotting**. For immunoblotting of whole-cell extracts, cells were lysed in 2× SDS-Lysis Buffer (20% Glycerin, 125 mM Tris pH 6.8, 4% SDS, 10% 2-Mercaptoethanol, 0.02% Bromophenol blue) and lysates were boiled at 95 °C for 10 min. Nuclear extracts were prepared using ice-cold hypotonic sucrose buffer (1% Triton X-100, 320 mM sucrose, 10 mM Hepes (pH 7.5), 5 mM MgCl2, 1 mM PMSF, 1× protease inhibitor cocktail, 1× phosphatase inhibitor cocktail). Cells were solubilized on ice with hypotonic buffer using a tissue grinder (Sigma) with 10–15 strokes per sample. After centrifugation, the supernatant was saved as cytoplasmic fraction and cell nuclei were washed twice with ice-cold sucrose wash buffer (320 mM sucrose, 10 mM Hepes (pH 7.5), 5 mM MgCl2, 1 mM PMSF, 1x protease inhibitor cocktail, 1x phosphatase inhibitor cocktail). To shear the DNA, nuclei were resuspended in ice-cold nuclear sonication buffer (50 mM Tris-HCl (pH 8.0), 150 mM NaCl, 1 mM EDTA (pH 8.0), 10% glycerol, 1 mM PMSF, 1× protease inhibitor cocktail, 1× phosphatase inhibitor cocktail) and sonicated twice for 10 s with a rest of 30 s on ice using a Branson 250 sonifier. Sheared DNA was removed by incubation with 250 U of benzonase (Sigma) for 30 min, rotating at 4 °C, followed by centrifugation. Nuclear extracts were transferred into fresh tubes and the protein concentration of cytoplasmic and nuclear lysates was determined using the Qubit Protein Assay Kit (Thermo Fisher Scientific). After the addition of 5× SDS-Lysis Buffer, nucleic and cytoplasmic extracts were boiled at 95 °C for 10 min. Whole-cell lysates, nucleic, and cytoplasmic extracts were separated by SDS-PAGE on 8% or 10% polyacrylamide gels and transferred onto a PVDF membrane (Merck) using a three-buffer semi-dry system. After 1 h of blocking in TBS with 5% dry milk at room temperature, blots were washed and incubated with anti-TET2 (MABE462, Merck, 1:1000), anti-IRF4 (sc-6059X, Santa Cruz, 1:5000), anti-EGR2 (sc-293195, Santa Cruz, 1:200), or anti-Actin (A2066, Sigma, 1:2000) overnight at 4 °C. After washing and incubation for 1 h at room temperature with a horseradish-peroxidase (HRP)-coupled secondary antibody (Goat anti-rabbit, P0448, Dako, 1:5000; Rabbit anti-goat, P0449, Dako, 1:5000; m-IgGk BP-HRP, sc-516102, Santa Cruz, 1:5000), blots were washed and proteins were visualized using the ECL Prime Western Blotting System (Sigma) and the Fusion Pulse imaging system from Vilber Lourmat. For re-probing, blots were stripped with 1× ReBlot Plus Mild Antibody Stripping Solution (Merck) for 15 min at room temperature.

**ChIP-seq**. ChIP-seq experiments were performed as described, with slight modifications[25]. Briefly, to crosslink protein–protein interactions, cells were initially crosslinked with 2 mM disuccinimidyl glutarate (DSG; Thermo Fisher Scientific) in PBS for 30 min at room temperature. Then, cells were further fixed with 1% formaldehyde for 10 min at room temperature and the reaction was quenched with glycine at a final concentration of 0.125 M. Chromatin was sheared using sonication (Branson Sonifier 250) to an average size of 250–500 bp. After sonication, 2.5 μg of antibody against PU.1 (Santa Cruz, sc-352X) and FLAG M2 (Sigma Aldrich, F3165) were added to the sonicated chromatin, followed by an incubation overnight at 4 °C. To pull down antibody–chromatin complexes, pre-blocked Protein G or A beads (GE Healthcare) were added to the lysate–antibody mix and rotated for 2 h at 4 °C. Beads were washed, the chromatin eluted, and subjected to reverse crosslinking by incubation overnight at 65 °C. Following RNase A and proteinase K treatment, DNA was purified using the Monarch PCR & DNA Cleanup kit (NEB). Sequencing libraries were prepared with the NEBNext Ultra II DNA Library Prep Kit for Illumina (NEB) according to manufacturer's instructions. The quality of dsDNA libraries was analyzed using the High Sensitivity D1000 ScreenTape Kit (Agilent) and concentrations were determined with the Qubit dsDNA HS Kit (Thermo Fisher Scientific). Libraries were single-end sequenced on a HiSeq3000 (Illumina). Sequencing libraries are listed in Supplementary Table 5.

**ATAC-seq**. ATAC-seq was carried out as described before[55]. Briefly, cells were harvested after 4 or 7 days of culture and washed. If viability (determined by trypan blue exclusion) was below than 50%, dead cells were first excluded using the Annexin V MicroBead Kit (Miltenyi Biotec) according with the manufacturer's instructions. Then, cells were treated with DNase I (Sigma) at a final concentration of 200 U/ml for 30 min at 37 °C in culture medium. After DNase I treatment, cells were washed, counted, and $5 \times 10^4$ viable cells were used per transposition reaction. Cells were resuspended in 50 μl of ATAC-RSB buffer (10 mM Tris-HCl (pH 7.4), 10 mM NaCl, 3 mM MgCl2) containing 0.1% NP-40, 0.1% Tween-20, and 1% digitonin (Promega) and lysed on ice for 3 min. After washing with ATAC-RSB buffer containing 0.1% Tween-20, nuclei were pelleted at 500g for 10 min at 4 °C in a fixed angle centrifuge. Pellets were resuspended in 50 μl of transposition mixture (25 μl 2x TD buffer, 2.5 μl transposase (100 nM final; Illumina), 16.5 μl PBS, 0.5 μl 1% digitonin, 0.5 μl 10% Tween-20, 5 μl H$_2$O), followed by incubation for 30 min at 37 °C with mixing (1000 rpm). DNA was then purified using the Monarch PCR & DNA Cleanup kit (NEB) according to manufacturer's instructions. Samples were eluted in 21 μl of elution buffer and 10 μl of purified DNA was used for library preparation using Nextera XT i7- and i5-index primers (Illumina) and 10 cycles of amplification. The amplified DNA was purified and size-selected using magnetic beads (Agencourt AMPure XP). ATAC libraries were analyzed using the High Sensitivity D1000 ScreenTape Kit (Agilent) and sequenced on a NextSeq550 (Illumina). Sequencing libraries are listed in Supplementary Table 6.

**RNA-seq library preparation**. Total cellular RNA was isolated from MO, MAC, and moDC (untreated, mock-, or siRNA-treated) using the RNeasy Mini Kit (Qiagen) according with manufacturer's instructions. RNA was quantified using the NanoDrop (peqLab) and the quality was assessed using the RNA ScreenTape Kit (Agilent). Generation of dsDNA libraries for Illumina sequencing was carried out using the ScriptSeq Complete Kit (Illumina) or the TruSeq Stranded Total RNA Kit (Illumina) according to manufacturer's instructions. The quality of dsDNA libraries was checked with the High Sensitivity D1000 ScreenTape Kit (Agilent) and concentrations were determined with the Qubit dsDNA HS Kit (Thermo Fisher Scientific). Sequencing was performed using the Illumina NextSeq550 sequencer and libraries are listed in Supplementary Table 7.

**WGBS analysis**. DNA methylation levels for individual CpGs across the genome were extracted from published WGBS bedGraph data (genome version hg19) of four MO and six moDC (summarized in Supplementary Table 8). Data sets were combined using the unionbedg function in bedtools v.2.27.1[56] and CpGs with a coverage of <5 reads for <2 MO or <3 moDC donors were excluded, leaving ~2.2 × 10$^7$ CpG residues. The density scatterplot shown in Fig. 1b was generated using the smoothScatter function in R, including the subset of CpGs on autosomes, not overlapping with repeats or blacklisted genomic regions (~1.1 × 10$^7$ CpG residues). To identify DMR, significance levels of DNA methylation changes at individual sites between MO and moDC replicates were determined using the Mann-Whitney test. DMR were defined as regions with at least two consecutive, significantly different CpGs showing ≥25% difference in DNA methylation between MO and moDC. For further analyses, DMR were filtered using the intersect function in bedtools by subtracting blacklisted genomic regions, removing regions overlapping more than 50% with repeats, restricting to autosomes and extending the region by 100-bp on either side. As random control regions with similar nucleotide distribution, we used a background region set generated by HOMER's findMotifsGenome.pl program, which was filtered as above (without extension). Genomic locations of DMR and random control regions were lifted from hg19 to hg38 using CrossMap 0.2.7[57], and further filtered for mappability ≥0.8 (annotated to peak regions from mappability tracks generated with the GEM package[58] using HOMER's annotatePeaks.pl program), leaving 7610 DMR demethylated and 121 DMR methylated in moDC, as well as 6098 control regions. Genomic distance distributions of MO and moDC CpG methylation values (as shown in Figs.1c, 3a,

d, 4f, 5f, 6a, 7b, and Supplementary Figs. 1d, 3a, 5g) were generated using the HOMER's annotatePeaks.pl program with parameters "-hist 1 -ghist" and WGBS bedGraph files with values shifted +1 to distinguish missing from 0 values. Available data points were averaged for each position and reshifted (−1) data were plotted along with a smooth spline using R. To distinguish early and late deme-thylating DMR (as in Fig. 3a, b and Supplementary Fig. 3a), we created five equal-sized bins from DMR ranked by their average methylation level in MO. The separation of DMR based on TF binding/chromatin accessibility was done using HOMER's mergePeaks program. We filtered for DMR overlap with TF peaks from data sets shown in Fig. 3c ("TF bound" DMR subset, 5095 regions), and splitted remaining DMR into subsets overlapping ATAC peaks ("open" DMR subset, 1270 regions) and non-overlapping DMR ("no peak" subset, 1245 regions) using bed-tools intersect program. Read coverage across sorted DMR (as shown in Fig. 3c) was calculated using HOMER's annotatePeaks.pl with parameters "-hist 25 -ghist" using the indicated data sets and plotted in R using the image function. The 5mC-spikes that may represent methylated CpGs protected from DNA demethylation by TF were identified in the indicated cell types using published WGBS data (sum-marized in Supplementary Table 8). Signal and coverage data were combined for each sample using the unionbedg function in bedtools v.2.27.1 and reduced to CpGs with more than 5 reads in at least half of the samples. The filtered, averaged methylation data were screened for increases in DNA methylation of >50% in neighboring CpGs (less than 100-bp apart), which were immediately followed by a decrease in DNA methylation of >50%. If the peak-CpG was followed by another CpG within a 6-bp window, their data were combined. The 5mC-spikes were filtered for mappability and repetitive elements. Overlaps between DMR, 5mC-spikes, and other peak sets were determined using the HOMER's mergePeaks program and presented as Venn diagram using the Venneuler package in R (Figs. 1e, 7d, e, and Supplementary Fig. 1e) or pie charts using the pie function of the plotrix package in R (Fig. 7g). Genomic locations of DMR, 5mC-spikes, or CpGs were determined using HOMER's annotatePeaks.pl program with the human GRCh38 genome and gene annotation from GENCODE 44 (release 27). Pie charts were plotted using the pie function of the plotrix package in R (Figs. 1d and 7c).

**EpiTyper analysis.** Methylation was quantified from mass spectra using the Epi-Typer software (v1.2, Agena). Methylation ratios for all samples and PCR ampli-cons underlying each figure are provided as Source Data file. Heatmaps presenting the CpG methylation levels across amplicons and individual replicate samples (Supplementary Fig. 1a), or averaged CpG methylation levels across amplicons and samples (Figs. 2b, 5g–j, and Supplementary Figs. 2c, e, 5h–k, 6a–d) were plotted using the heatmap2 function of the gplots package in R. Boxplots of individual CpGs across samples were plotted in R using the ggplot2 package. Scatterplots comparing EpiTYPER and WGBS methylation ratios for randomly selected moDC or DMR methylated in moDC (Supplementary Fig. 1b) were plotted in R using the ggplot2 package.

**5hmC capture-seq.** Reads (single-end) were aligned to the human genome (GRCh38/hg38) using bowtie2[59] in very sensitive mode, keeping only reads that map to a single unique genomic location for further analysis (MAPQ > 10). Initial quality control was performed by calculating the fraction of reads in peaks (FRIP, summarized in Supplementary Table 4) by running HOMER's[60] (v4.9) findPeaks program in "histone" mode using default parameters and the matching background data set (DNA input). For further analyses, 5hmC capture-seq peaks were called using HOMER's findPeaks program with "-region -size 250 -L 0 -F 5 -minDist 350 -fdr 0.00001 -ntagThreshold 10". Peak sets of the three time points (MO, DC18h, and DC7d) were merged using bedtools merge program and filtered by subtracting blacklisted genomic regions[61] and by filtering out regions with a mappability of <0.8. Density scatterplots shown in Supplementary Fig. 1c were generated using the smoothScatter function in R, including the subset of peaks located on autosomes, not overlapping with repeats. Read coverage across individual peak sets (as shown in Fig. 1g) was calculated using HOMER's annotatePeaks.pl with parameters "-hist 25 -ghist" using merged replicate 5hmC capture-seq data sets and plotted in R using the image function. Average coverage data and 95% confidence intervals were calculated in R and the ggplot2 package was used to draw histograms shown in Fig. 1f.

**RNA-seq analysis.** Sequencing reads were mapped to the human genome using STAR v2.5.3a[62]. The human GRCh38 genome index incorporated gene annotation from GENCODE 44 (release 27) was used to aid in spliced alignment. Tables of raw uniquely mapped read counts per human gene were generated during mapping using the built-in --quantMode GeneCounts option in STAR. Differential expression analysis was carried out on raw gene counts using edgeR 3.20.9[63] in R (v3.4.3). Pairwise comparisons of indicated data sets were done using the quasi-likelihood test. Z scores were calculated using the scale function in R. Heatmaps of differentially expressed genes shown in Figs. 2c, 4b, and 5b used log2-transformed, batch-corrected, normalized, and scaled CPM (counts per million) data, and were generated using the heatmap.2 function of the gplots package in R. Dimensionality reductions based on the tSNE algorithm (Supplementary Figs. 2d and 5c) were done using the Rtsne package and visualized using the ggplot2 package in R. Gene set enrichment analyses as shown in Figs. 2d, 4c, 5d, and Supplementary Fig. 5d

were performed using the function fry of the limma package[64] (v3.34.9) in R. For the comparison of gene expression levels between selected gene sets as shown in Supplementary Figs. 2f, 3c, 4c, and 5e, normalized and batch-corrected expression data were corrected for transcript length and plotted for selected genes using the ggplot2 (v3.1.0) package in R. Statistically significant enriched Gene Ontology terms were identified using Metascape[65], and barplots of significance levels shown in Figs. 4d and 5c were generated in R.

**ATAC-seq analysis.** Reads (paired-end) were aligned to the human genome (GRCh38/hg38) using bowtie2 in very sensitive and no-discordant modes, keeping only reads that map to a single unique genomic location for further analysis (MAPQ > 10). Read positions were adjusted to move the ends proximal to the Tn5-binding site (for reads on the positive strand, the start is shifted +4 bp and its partner reads start −5 bp, for reads on the negative strand, the start is shifted −5 bp and its partner reads start +4 bp). Initial quality control was performed by calculating the FRIP (summarized in Supplementary Table 6 and for published data in Supplementary Table 9) by running HOMER's findPeaks program using parameters "-region -size 150". ATAC-seq peak regions were called by combining two different approaches: The basic peak region set was called using HOMER's findPeaks program in "region" mode using parameters "-size 150 -minDist 250 -L 2 -fdr 0.00001" to identify regions of variable length by stitching nucleosome-size peaks. To exclude shallow peak regions, only those were kept that overlapped a second peak set that were generated in "factor" mode using parameters "-size 250 -minDist 250 -L 2 -fdr 0.00001" to identify focal peaks. Statistically significant differences in read counts across peaks between sets of replicate ATAC-seq experiments were determined with quantile (0.95) nor-malization and GC correction using edgeR 3.20.9[63] with the cqn package[66] in R (v1.24.0). Read coverage across individual peaks sets (as shown in Figs. 1g, 4e, and 5e) was calculated using HOMER's annotatePeaks.pl with parameters "-hist 25 -ghist" using merged replicate ATAC-seq data sets and plotted in R using the image function. Average coverage data and 95% confidence intervals were calculated in R and the ggplot2 package was used to draw histograms (shown in Figs. 1f, 3e, and Supple-mentary Fig. 3f). To generate scatterplots comparing ATAC-seq data for siTET2- and siCTRL-transfected cells shown in Fig. 2e, reads were counted across the indicated peak sets using HOMER's annotatePeaks program. Scatterplots were drawn in R using the ggplot2 package and corresponding correlation coefficients were calculated in R.

**ChIP-seq analysis.** Reads (single-end) were aligned to the human genome (GRCh38/hg38) using bowtie2[59] in very sensitive mode, keeping only reads that map to a single unique genomic location for further analysis (MAPQ > 10). Initial quality control was performed by calculating the FRIP (summarized in Supple-mentary Table 5 and for published data in Supplementary Table 10) by running HOMER's[60] (v4.9) findPeaks program in "factor" or "histone" mode using default parameters and the appropriate matching background data set (either ChIP input, genomic DNA, or control ChIP). For further analyses, chromosome scaffolds were removed. TF ChIP-seq peaks were called using HOMER's findPeaks program in "factor" mode with "-fdr 0.00001 to identify focal peaks. Peak sets were filtered by subtracting blacklisted genomic regions[61], and by filtering out regions with a mappability of <0.8. The latter was annotated to peak regions from mappability tracks generated with the GEM package[58] using HOMER's annotatePeaks.pl.

**Motif analysis.** De novo motif discovery in peaks or regions (e.g., as shown in Figs. 3b, 4e, 5e, and Supplementary Fig. 3b, e, f) was performed with HOMER's findMotifsGenome.pl program and parameters "-len 7,8,9,10,11,12,13,14 -h". For searches in ChIP-seq peaks we used a 200 bp, peak-centered window, while for DMR or differential ATAC regions the given region sizes were used. De novo motifs were further filtered using HOMER's compareMotifs.pl and parameters "-reduceThresh .75 -matchThresh .6 -pvalue 1e-12 -info 1.5". Motif log-odds score for EGR, ETS, and STAT motifs across DMR sets were calculated using HOMER's annotatePeaks program. Distributions of motif scores (as shown in Fig. 3g) were visualized using the beanplot package in R. To determine peak-wise motif co-association, we first performed a known motif search using HOMER's findMo-tifsGenome.pl across DMR sets and determined the list of known motifs over-lapping the previously determined de novo motif classes (e.g., Ebox, GATA, or RUNX). All listed motifs were then counted in peak regions using HOMER's annotatePeaks.pl with parameters "-mknown.motifs -matrixMinDist 4 -nogene -noann -nmotifs". Motif overlap in each individual peak was then reduced to motif class overlap (using the filtered known motif list), which was counted as positive for a particular class if one of the class matching known motifs was present, or negative if none was present. The count table was then used to generate a motif co-occurrence matrix and to calculate node sizes and edges' width (each represented as % of all peaks). Networks of motif co-association were generated in R using the igraph package (as shown in Fig. 3f). To improve the visualization, colors of individual nodes were edited in Adobe Illustrator.

**Generation of read coverage tracks.** HOMER was used to generate sequencing-depth normalized bedGraph/bigWig files of ChIP-seq and ATAC-seq data (using standard parameters for ChIP and a fixed fragment length of 65 bp for ATAC).

ATAC-seq tracks were quantile normalized based on the top 5000 peaks across all samples. BigWigs from replicate data sets were averaged using the program bigWigMerge[67] and dividing the count data by the number of samples. Resulting bedGraph files were converted to BigWig using the program bed-GraphToBigWig[67]. Tracks were visualized using the IGV browser[68]. Selected regions (as shown in Figs. 1h, 2b, 5g–j, 6b, 7h, and Supplementary Figs. 1f, 2c, e, 5h–k, 6a–d) were exported in svg format and formatted in Adobe Illustrator.

**Reporting summary**. Further information on research design is available in the Nature Research Reporting Summary linked to this article.

## Data availability

The data that support this study are available from the corresponding author upon reasonable request. The EGA study accession number for the NGS raw data is EGAS00001004784. Processed data files (bigwig tracks and peak files for ATAC, ChIP, and 5mC-capture data and read count tables for RNA-seq data) are deposited with ArrayExpress accession codes E-MTAB-9926, E-MTAB-9927, E-MTAB-9928, E-MTAB-9929. Source data are provided with this paper.

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

## Acknowledgements

We thank Alain Pacis and Luis B. Barreiro (University of Chicago) for sharing data before publication. This study makes use of data generated by the Blueprint Consortium. A full list of the investigators who contributed to the generation of the data is available from www.blueprint-epigenome.eu. Funding for the project was provided by the European Union's Seventh Framework Programme (FP7/2007-2013) under grant agreement no. 282510 BLUEPRINT. ChIP- and RNA-sequencing were conducted at the biomedical sequencing facility (BSF) of the CeMM (Vienna, Austria), at the KFB (Kompetenzzentrum Fluoreszente Bioanalytik, Regensburg, Germany) and the NGS Core of the Regensburg Center for Interventional Immunology (RCI, University Regensburg and University Medical Center Regensburg, Germany). This study was funded by a grant of the DFG to M.R. (RE 1310/21).

## Author contributions

M.R designed the study; K.M. performed most experiments with contributions from S.S., J.M., D.G., C.K., J.R., J.W., and C.G.; M.R. analyzed sequencing data with help from K.M. and S.S.; M.R. and K.M. wrote the original draft with contributions from all authors.

## Funding

## Competing interests

The authors declare no competing interests.
