## [Peer Review File. · Nature Communications]

Reviewer #1 (Remarks to the Author):

The study is relevant and well executed with nice functional profiling (knock-down, FLAG-TF ChIP, multiple dataset integration). It builds on a large body of previous work by the same group in the same cell model, with a shift from descriptive mapping of DMRs to functional evaluation. I have comments about specific analysis steps, data availability and presentation.

1. Please load the data to GEO and provide a reviewer password so that the completeness of the dataset can be verified. This will enable future studies to incorporate their data in future studies and meta-analyses.
2. Reference missing on line 70: "... through its ability to recruit TET2." The other possibility is that this is somehow the start of the summary of the current study. It is a little odd to have half the introductions summarising the current study with no references or other confirmatory data introduced. Best to shorten this to 2-3 sentences introducing your current paper and then go into the results.
3. There is no reference to Figure 1C in text.
4. The authors should clearly state in the first paragraph of the results how many Ns they had in the sequencing experiments for each epigenetic mark.
5. It is unclear why 'DMR' is taken to mean 'demethylation region'. Why are regions that lose DNA methylation during differentiation 'DC-specific' and those that gain DNA methylation 'MO-specific'? The authors outline what they mean at the beginning (line 106) so it's fine, but I have never seen it described that way before in papers – perhaps it is when you enter the question from the TET perspective. Why not: hypermethylated in DC, hypomethylated in DC, which to me is more informative?
6. For identifying DMRs, it is important to know coverage of the sequencing. Can the authors please let us know what the cutoff reads were for using a CpG datapoint in their analysis? This refers to the initial DMR analysis (lines 692-704), not the spikes where a coverage of 5 is stated (line 727).
7. Figure 2 and line 145, please state how many total ATAC peaks were called and how many are affected by TET2 depletion.
8. Lines 187-190. Why the assumption that 'demethylation has already been initiated'? Could not these regions have gained methylation from progenitor to monocyte?
9. Figure 3B, should "iDC DMR bins" say "MO DMR bins"?
10. It is not clear from Figure 3B or the legend what 'motif enrichment log2' means. Is it the fold change relative to background or percentage abundance? PU.1 does not look to be highly enriched from the heatmap.
11. It is also not clear how many DMRs are in each MO bin in Figure 3.
12. The 'open' pattern group in Figure 3E is interesting. Have you checked the motif enrichment at ATAC peaks adjacent to the 'open' DMR ATAC peak?
13. Sometimes the authors are very selective in which genes they show and how they show them. This is done to make the story 'cleaner' I assume, but the cherry-picking can make it hard to understand what is actually happening. For example: Figure 2B, Figure 4B. For 4B, are these the only genes different when IRF4 is depleted? Or a fraction? Making the two heatmaps separate makes it look perfect, but what is the actual expression of these genes in IRF4 depleted cells and macrophages (within 10%, 50%)? I agree that it's fine to show polished version in main figures, but the supplementary Figure 4 should plot a boxplot or bar, or at least put these 20 odd genes in context of all the genes affected by IRF4.
14. Figure 4D, it would be useful to see if these ATAC peaks are also missing/gained in macrophages. Similarly how it was shown for RNA expression in Figure 4B.
15. Line 303: 'loose' should be 'lose'.
16. Figure 4E is missing the line for IRF4 knockdown. Line 305-308, again I am not sure how the authors can know that these regions were already being actively demethylated in monocytes. Could be going up from progenitor (zero) to monocyte (50%), could equally be passive demethylation.
17. Supplementary figure 5C, color for iDC siRNA control does not match between legend and plot.
18. EGR2 is blocking differentiation at level of RNA (Supplementary Figure 5C). Is the same pattern true for open chromatin?
19. Figure 6A. I do not see the pattern the authors are claiming. Both EGR2 motif and AP1 motif

regions show demethylation during differentiation. Perhaps I missed the point here. Is there another way to illustrate this? I see the dip at AP-1 regions in monocytes, but how does this go with the statement "generally remain methylated". This statement implies they do not demethylate during differentiation.

20. Line 364-368. I do not follow this logic: "Hence, demethylation of CpGs around these binding sites only proceeds when the binding site is occupied." The CpG sites at these two different motif regions (EGR2 and AP-1) are presumably different regions of the genome. Where is the evidence that the binding site must be occupied? I get the second part of the sentence that the occupying factor needs to recruit demethylation machinery.

21. I also get lost in lines 405-417. In Figure 6A the authors make the claim that EGR2 blocks demethylation. Then in lines 405-417 they claim EGR2 does recruit TET2 to induce demethylation. Again I may have missed something.

22. Line 431 – please add reference.

23. Methods cell culture – while the authors reference their previous studies (2 in fact), it would still be useful for the reader to know briefly how monocytes were isolated: percol? Negative beads? Positive bead?

Reviewer #2 (Remarks to the Author):

This is a complex paper that has potential interest. The focus is on the mechanism underlying global DNA demethylation during monocyte differentiation. However, the definitive statement in the title: "The epigenetic pioneer EGR2 initiates DNA demethylation in differentiating monocytes at both stable and transient binding sites" is not proven by the data.

Specific comments

1. Gene names and protein names should be consistent with current nomenclature and format (CEBPA, KLF4, TCFP2L1, TET2 for protein)

2. The rationale for the use of the GM-CSF/IL4 system is not provided. It is commonly recognized that monocyte-derived APC are distinct from classical DC; MDDC is the more common abbreviation. It is unclear whether IL4 is necessary or relevant in this system. GM-CSF stimulated monocytes are also considered as novel inflammatory macrophages (e.g. PMID:27525438) and CCL17 (which is one of the induced genes in Figure 2b) is a pro-inflammatory effector. The paper is really about monocyte differentiation and the end point is not that relevant. If iDC is used as an abbreviation, it should be better defined and justified with reference to the literature.

3. EGR2 is also induced by IFN γ in monocytes and is rapidly induced during monocytic differentiation of THP-1 cells in response to PMA. It is expressed by MDM as well as MDDC. So, it is not clear that it is specific to "DC-related transcriptional programs". As background to its proposed role as a pioneer it would be useful to know the time course of induction and nuclear accumulation in monocytes treated with GM-CSF/IL4.

4. It was not clear to me how "example regions" in Figure 1h and Supp Fig 1f were chosen and what they were meant to show. Needs justification and more description in the text. Perhaps worth noting from other data that CLEC10A and DNASE1L3 are specifically induced by GM-CSF but not by M-CSF. SLAMF1 and MAF on the other hand are not specific. The previous study (Ref 9) used CCL13 as an example; it is also strongly GM-CSF inducible.

5. Figure 1 is an analysis of methylation states from published data. The source publications and their context should be cited in the main text. It is a misnomer to refer to DC-specific DMR. They are relatively demethylated only by comparison to monocytes. Many are likely also demethylated in response to M-CSF and/or in other cell types. It would actually be useful to identify the subset of DMR that are specific to MDDC versus MDM (grown in M-CSF).

6. "In line with the unidirectional DNA methylation turnover, regions that lose chromatin accessibility (as detected by ATAC sequencing) maintain their DNA methylation state while those that gain accessibility are demethylated (Supplementary Figure 1d)" (Spelling error and would be better in past tense).

7. Figure 2a indicates rather poor viability in the siRNA treatment and Figure S2b is too small and insufficient resolution to document unchanged morphology.

8. Paragraph on bottom of page 5 has multiple syntax errors and needs to be re-written. the use of words like "few" and "mild" creates ambiguity; they mean different things to different people. I think the take home message is that the Tet2 knockdown selectively impacted only a small subset of the genes that were induced during differentiation. It is stated that "A large fraction of downregulated genes (22 in total) were normally induced during iDC differentiation (Figure 2b & S2d), indicating a mild but reproducible transcriptional effect of the TET2 knock-down". Large and mild and reproducible are not defined. How reproducible was this fraction in multiple replicates? What is the likelihood of this occurring by chance based upon the number of genes that are induced during monocyte differentiation (which is quite large). What proportion of genes that were induced were not impacted by TET2 knockdown.

9. I may have missed the logic, but why is the set of 3 example genes in Figure 2d different from the set of 22 differentially regulated genes in Figure 2b.

10. p6 states "Gene expression data suggested that IRF4 (sic) and EGR2 are transcriptionally activated in iDC (Supplementary Figure 3b), qualifying them as candidate factors driving de novo demethylation processes." It was not immediately clear from the legend or text where these data came from. In published data, IRF4 is induced in MDDC but not MDM; EGR2 is not specific to the GM-CSF treated cells.

11. p6. Text states "Consistent with the slight loss of chromatin accessibility across DMR (Figure 2c), we observed a slight reduction of ATAC signals across PU.1 and AP1 motifs at DMR in TET2 knock-down samples (Supplementary Figure 3c)". This does not seem compelling from the Figures. Can "slight" be quantitated and what is the FDR if one looks at a comparable size permuted set.

12. The IRF4 knockdown experiment is not described sufficiently. It is not clear from the text or methods when the siRNA treatment was performed (on monocytes prior to differentiation?), how long afterwards viability was measured and by what method. Suppl Fig4B is not sufficient resolution to support the finding of "drastically altered cell morphology". Suggest that since it is supplementary, larger images and a comparison with MDM grown in M-CSF would be more convincing. The data in Figure 4b/c is difficult to interpret and scaled data gives a misleading impression of magnitude. One might conclude from this figure that MDM lack class II MHC gene expression, which is certainly not the case in other datasets. The Z score is not defined. I could not find an accession number for the data. Supplementary Table 7 shows Acc XXX. For greater utility to allow others to replicate the analysis all the RNA-seq processed data (CPM or RKPM) should be available as Supplementary data in an Excel data sheet. The GO analysis gives little additional insight and could be removed. The set of genes apparently up-regulated (or not down-regulated) in the IRF4 knockdown is not discussed and they give clear insights into possible mechanism, including transcription factors (PPARG, IRF8, CEBPB) and possible autocrine growth factor (CSF1).

13. Same comments relate to the EGR2 knockdown but even more so. As presented, the effect is difficult to interpret There is no obligate causal link between EGR2 knockdown and gene expression or methylation changes. How should we interpret an effect of the siRNA if 75% of cells are lost. Suppl Figure 5B is inadequate to describe the morphology of surviving cells. Are the EGR2 knockdown cells undergoing apoptosis? Is this simply an analysis of the selected subset of cells (monocytes or MDDC?) that survive the EGR2 knockdown and perhaps did not express EGR2 anyway? Is the expression of EGR2 heterogeneous in unselected population. This interpretation is supported by the observation that "Genes upregulated in MO cultures upon EGR2 knock-down also included a number of T cell specific marker genes, suggesting that contaminating cells were unaffected (and hence enriched) by the EGR2 knock-down". The set of genes that is apparently up-regulated by the EGR2 knockdown looks like a monocyte signature (IRF8, CEBPB, S100A8/9, CD14) and again is not discussed. It may be that undifferentiated monocytes are the only cells that survive. It may still be possible to infer a relationship between EGR2 binding and demethylation but with significant caveats.

14. p11. "The almost complete resistance to demethylation suggested that the demethylation machinery does not have access to the protected CpGs. Hence, demethylation of CpGs around these binding sites only proceeds when the binding site is occupied, and the occupying factor must be critical for recruiting the demethylation machinery." The logic is not clear to me and the Figure 6A is at

insufficient size, resolution and color contrast to decipher the intended message.

15. The discussion provides an unconvincing explanation for the lack of effect of the TET2 knockdown on demethylation. Their previous study concluded that TET2 was the major demethylase in monocytes and TET3 was less highly-expressed. There is a recent literature on TET3 in monocytes and it is certainly detectable at the mRNA level in published data. Could redundancy be an alternative explanation? The authors might comment on detection of TET3 in their own RNA-seq data.

16. The discussion finishes on a bit of an afterthought.

Response to reviewer's comments:

We would like to thank the reviewers for the time and effort they have invested in reviewing our manuscript. In response to their valuable comments and recommendations, we have acquired additional data to further support our findings, including new ATAC data, provide additional analyses and extensively revised the data presentation. We feel that the new data and analyses significantly strengthen our conclusions. We hope the reviewers now agree that our findings make a significant and important contribution to our understanding of active DNA methylation turnover.

All changes in the main or supplemental text are indicated by red lettering. Changes in figures include:

- General: Exchanged labels; “DC” or “iDC” to “moDC”; “MO specific DMR” to “DMR methylated in moDC”; “iDC specific DMR” to “DMR demethylated in moDC”.
- Figure 1h: introduced transparent green boxes to highlight changes in 5hmC and ATAC signals across remodelled DMR.
- Figure 2: reordered panels to first show validation of TET2 knock-down effect on DNA methylation level, before presenting effects on gene expression and chromatin accessibility.
- Figure 2a: Included viability data for untransfected moDC, showing that the electroporation does not significantly affect the viability.
- Figure 2c: Show heatmap of all differentially expressed genes (instead of subset).
- Figure 2d: Added new GSEA showing that genes downregulated by TET2 are significantly enriched in moDC (compared to MO).
- Figure 4b: Corrected heatmap (the original presentation was accidentally based on a more relaxed filtering) and removed gene labels to avoid the impression of “cherry picking”.
- Figure 4c: Added new GSEA analysis demonstrating that the gene set representing IRF knockdown-induced genes is enriched in macrophage differentiation-associated genes.
- Figure 4d: Revised representation of gene-ontology terms (using the correctly filtered expression data).
- Figure 4e: Added new ATAC data for MO-derived MAC which resemble the IRF4 kd ATAC profiles across differentially accessible regions..
- Figure 5b: Corrected heatmap (the original presentation was accidentally based on a wrong filtering) and removed gene labels to avoid the impression of “cherry picking”.
- Figure 5c: Removed GO term analyses of EGR2 knockdown-induced genes and corrected analysis for genes downregulated upon EGR2 knock-down.
- Figure 5d: Added new GSEA analysis demonstrating that the gene set representing moDC induced genes is depleted in EGR2 knock-down cells (compared to siCTRL-treated cells), while genes upregulated in MO (compared to moDC) are enriched in EGR2 knock-down cells (compared to siCTRL-treated cells).
- Figure 5e: Added new ATAC data for MO which resembles the EGR2 kd ATAC profiles across differentially accessible regions.
- Figure 6a: Added additional inlays focusing into the DNA methylation levels across the EGR/AP1 motifs.
- Supplementary Figure 1: Rearranged Figure panels and increased their size for better resolution.
- Supplementary Figure 1f: introduced transparent green boxes to highlight changes in 5hmC and ATAC signals across remodelled DMR.
- Supplementary Figure 2: rearranged Figure panels as part of the rewritten results section.
- Supplementary Figure 2b: Added new microscopy images with higher resolution, including both untransfected and control-transfected cells.
- Supplementary Figure 2d: Added new tSNE-plot showing the distribution of RNA expression data sets.
- Supplementary Figure 2f: Added bar plots showing RNA expression of TET family members based on our RNA-seq data.
- Supplementary Figure 3a: new analysis supporting the progressive demethylation in monocytes.
- Supplementary Figure 3c: new Western blot analysis demonstrating the induction and presence of IRF4 and EGR2 in nuclear extracts.
- Supplementary Figure 3f: new analyses showing differential motif enrichment in central and adjacent accessible chromatin of the “open” subgroup of DMRs.
- Supplementary Figure 4b: Added new microscopy images with higher resolution.

- Supplementary Figure 4c: Added bar plots showing RNA expression of differentially expressed genes based on our RNA-seq data.
- Supplementary Figure 5b: Added new microscopy images with higher resolution.
- Supplementary Figure 5c: Fixed color transparency issue.
- Supplementary Figure 5d: Added new GSEA analysis demonstrating that the gene set LYMPHOCYTE_DIFFERENTIATION is enriched in EGR2 knockdown-induced genes.
- Supplementary Figure 5e: Added new expression data analysis showing EGR2 kd induced changes in myeloid or lymphoid transcription factors.
- Supplementary Figure 5f: Added new IGV genome browser tracks comparing changes in chromatin accessibility in control and knockdown cells.
- Supplementary Figure 6: Rearranged Figure panels and increased their size for better resolution.

Point-by-point response:

Reviewer #1 (Remarks to the Author):

The study is relevant and well executed with nice functional profiling (knock-down, FLAG-TF ChIP, multiple dataset integration). It builds on a large body of previous work by the same group in the same cell model, with a shift from descriptive mapping of DMRs to functional evaluation. I have comments about specific analysis steps, data availability and presentation.

1. Please load the data to GEO and provide a reviewer password so that the completeness of the dataset can be verified. This will enable future studies to incorporate their data in future studies and meta-analyses.

This was our first submission of human raw data to the EGA and the paperwork took longer than expected. However, all data has now been submitted and the processed data will be freely accessible. Accession number of the entire study is EGAS00001004784. Processed data files (bigwig tracks and peak files for ATAC, ChIP and 5mC-capture data and read count tables for RNA-seq data) are deposited with ArrayExpress (accession numbers: E-MTAB-9926, E-MTAB-9927, E-MTAB-9928, E-MTAB-9929). Reviewer links to the processed data files are:

<https://www.ebi.ac.uk/arrayexpress/experiments/E-MTAB-9926>

Username: Reviewer_E-MTAB-9926

Password: TW0wkoxp

<https://www.ebi.ac.uk/arrayexpress/experiments/E-MTAB-9927>

Username: Reviewer_E-MTAB-9927

Password: jhedbffb

<https://www.ebi.ac.uk/arrayexpress/experiments/E-MTAB-9928>

Username: Reviewer_E-MTAB-9928

Password: aadhorgw

<https://www.ebi.ac.uk/arrayexpress/experiments/E-MTAB-9929>

Username: Reviewer_E-MTAB-9929

Password: gvupihgf

2. Reference missing on line 70: "... through its ability to recruit TET2." The other possibility is that this is somehow the start of the summary of the current study. It is a little odd to have half the introductions summarising the current study with no references or other confirmatory data introduced. Best to shorten this to 2-3 sentences introducing your current paper and then go into the results.

In response to both reviewers comments we re-written parts of the introduction and have now added some additional information regarding the culture system and changed the abbreviation iDC to moDC which is more commonly used in the field. We have also shortened the summary of our findings to key points.

3. There is no reference to Figure 1C in text.

The corresponding reference has been added.

4. The authors should clearly state in the first paragraph of the results how many Ns they had in the sequencing experiments for each epigenetic mark.

In the original version most Ns were reported in Figures or Figure legends (which is also requested by the Journal). We have gone through each Figure and Figure legend to make sure that Ns are given for each experiment and corrected a typo in the 5hmC data in Figure 1 which correspond to two (and not three) independent replicates. In addition, our supplement contains the complete list of NGS samples.

5. It is unclear why 'DMR' is taken to mean 'demethylation region'. Why are regions that lose DNA methylation during differentiation 'DC-specific' and those that gain DNA methylation 'MO-specific'? The authors outline what they mean at the beginning (line 106) so it's fine, but I have never seen it described that way before in papers – perhaps it is when you enter the question from the TET perspective. Why not: hypermethylated in DC, hypomethylated in DC, which to me is more informative?

DMR (at its first usage, page 5) was originally defined as “differentially methylated region”. Apparently, the separation between moDC-specific and MO-specific DMR was misleading and we have now rephrased our definitions to avoid misunderstandings. We wanted to keep the terms demethylated/methylated to reflect the processing rather than using hyper and hypo, but we hope that the definitions (DMR methylated or demethylated in moDC) now make sense.

6. For identifying DMRs, it is important to know coverage of the sequencing. Can the authors please let us know what the cutoff reads were for using a CpG datapoint in their analysis? This refers to the initial DMR analysis (lines 692-704), not the spikes where a coverage of 5 is stated (line 727).

We used a coverage cut-off of 5 for all published WGBS data sets. We have now added this information in the Methods section.

7. Figure 2 and line 145, please state how many total ATAC peaks were called and how many are affected by TET2 depletion.

This information has now been added in the main text.

8. Lines 187-190. Why the assumption that 'demethylation has already been initiated'? Could not these regions have gained methylation from progenitor to monocyte?

We agree with the reviewer – in principle, it is indeed possible that these regions gained methylation from progenitor to monocyte. In response to this comment and to systematically test our assumption, we compared CpG methylation levels of MO and moDC with published data for CD34 progenitor cells. As it is clear from the new Supplementary Figure 3a, average DNA methylation levels exactly follow the assumed pattern with highest levels in early progenitor cells, followed by MO and moDC. This data clearly supports our assumption and we rephrased the corresponding paragraph in the main text and included the data presented in Supplementary Figure 3a.

9. Figure 3B, should “iDC DMR bins” say “MO DMR bins”?

The typo has now been corrected.

10. It is not clear from Figure 3B or the legend what 'motif enrichment log2' means. Is it the fold change relative to background or percentage abundance? PU.1 does not look to be highly enriched from the heatmap.

The motif enrichment corresponds to the ratio of peak/region-associated motifs versus background motifs. We have now added this information in the revised Figure legend. The PU.1 position weight matrix produces the highest background; hence its enrichment over background appears weaker than the for the other factors. Nevertheless, its enrichment is highly significantly with p values ranging from 10^{-94} (bin1) to 10^{-45} (bin 5). For interested readers, this data will be available in the source data file.

11. It is also not clear how many DMRs are in each MO bin in Figure 3.

Quantiles are per definition equally sized; hence each bin (corresponding to a quintile) contains ~1.5K DMR (of our original 7.6K DMR set). To make this clearer, we added this information in the legend to Figure 3

12. The 'open' pattern group in Figure 3E is interesting. Have you checked the motif enrichment at ATAC peaks adjacent to the 'open' DMR ATAC peak?

Good suggestion. We have now added motif enrichment analyses (as in Figure 3B) and show that the central accessible region is enriched for AP1, EGR2 and STAT motifs, while the adjacent accessible regions are more enriched for PU.1 and ETS:IRF composite sites. This distribution is in line with our central finding that EGR2 is involved in the active demethylation.

13. Sometimes the authors are very selective in which genes they show and how they show them. This is done to make the story 'cleaner' I assume, but the cherry-picking can make it hard to understand what is actually happening. For example: Figure 2B, Figure 4B. For 4B, are these the only genes different when IRF4 is depleted? Or a fraction? Making the two heatmaps separate makes it look perfect, but what is the actual expression of these genes in IRF4 depleted cells and macrophages (within 10%, 50%)? I agree that it's fine to show polished version in main figures, but the supplementary Figure 4 should plot a boxplot or bar, or at least put these 20 odd genes in context of all the genes affected by IRF4.

In response to this comment and to avoid the impression of "cherry-picking", we have removed gene labels in heatmaps representing RNA seq data in Fig.2,4 and Fig.5. Here, originally only hand-picked selections of genes were labeled (mainly because we cannot label several hundred genes). The revised heatmaps now represent all differentially expressed genes in the presented comparisons (using the given thresholds).

Data sets in Figure 2b and 4b were originally separated because they derived from different studies, which makes batch-correction more difficult and generally increases noise in the analysis. Nevertheless, we have now combined experiments for presentation purposes and as seen in the new Figures 4b, the patterns largely remain the same. To allow for a side-by-side comparison of selected individual genes, we also added bar plots of normalized rpkm levels for selected genes across experiments and conditions. To further support the observed phenotypes, we have also now added gene set enrichment analyses (new panels in Figures 2,4,5) further supporting our claims.

14. Figure 4D, it would be useful to see if these ATAC peaks are also missing/gained in macrophages. Similarly how it was shown for RNA expression in Figure 4B.

We agree that this would be an interesting data set and now provide the corresponding ATAC-seq data for monocyte-derived macrophages (MAC). As seen in the revised Figure 4e, the ATAC-seq profile of MAC resembles IRF4 kd cells, which further strengthens our conclusions.

15. Line 303: 'loose' should be 'lose'.

The typo has now been corrected.

16. Figure 4E is missing the line for IRF4 knockdown. Line 305-308, again I am not sure how the authors can know that these regions were already being actively demethylated in monocytes. Could be going up from progenitor (zero) to monocyte (50%), could equally be passive demethylation.

In Figure 4e (now 4f) we are plotting MO and moDC data onto differentially accessible sites identified in the IRF4 kd experiment. We have added this information to the Figure to make this clearer.

As to the second issue: we are focusing on DNA methylation turnover during MO differentiation, which excludes passive demethylation. The point here is that the methylation state in MO already indicates demethylation at some earlier stage. Our assumption that demethylation at these sites is progressive (as opposed to *de novo*) is now also supported by the additional data shown in Supplementary Figure 3a (also see comment 8 above), indicating that methylation levels of progenitor cells are consistently higher than those of MO across DMR demethylated in moDC.

17. Supplementary figure 5C, color for iDC siRNA control does not match between legend and plot.

For some unknown reason the dot in the legend was shown with 50% transparency. This has been corrected and the colors are matching now.

18. EGR2 is blocking differentiation at level of RNA (Supplementary Figure 5C). Is the same pattern true for open chromatin?

In response to both reviewers comments we have added new analyses of the data related to the EGR2 knock-down experiments. This also includes the additional comparison of the open chromatin in EGR2 knock-down cells with monocytes. The revised Figure 5d clearly shows that the patterns in EGR2 knock-down cells across differentially accessible regions between control and EGR2 knock-down cells resemble the open chromatin in MO. This is particularly evident for sites that would normally gain accessibility during differentiation but remain inaccessible in the EGR2 knock-down.

19. Figure 6A. I do not see the pattern the authors are claiming. Both EGR2 motif and AP1 motif regions show demethylation during differentiation. Perhaps I missed the point here. Is there another way to illustrate this? I see the dip at AP-1 regions in monocytes, but how does this go with the statement "generally remain methylated". This statement implies they do not demethylate during differentiation.

Figure 6A is actually meant to show a key analysis of this paper and we apologize that it wasn't illustrated clear enough. To improve the presentation, we slightly altered the coloring (to better distinguish MO and moDC data sets) and added a smaller inlay figure zooming into the graph and showing DNA methylation values across the transcription factor motif sequences only. We hope that the lack of demethylation at the indicated CpGs within the EGR2 motif is now obvious. This "demethylation protection" is restricted to the CpGs overlapping the 10-12 bp binding site, which may have not been clearly visible in the original images.

Our point here is that, while demethylation clearly occurs in the area around the motif, CpGs within the motif itself are not demethylated. This type of pattern (which we later call CpG spikes) suggests that CpGs overlapping the binding site of EGR2 are protected from TET2-mediated demethylation (while demethylation clearly occurs at surrounding CpGs). The obvious explanation for this protection is that the binding site is occupied by the transcription factor EGR2.

This binding of EGR2 at consensus sites is confirmed by our ChIP-seq data in the 'TF peak' group of DMRs, and this is in line with published observations that EGR2 can bind its motif regardless of its DNA methylation state. Following this line of argument, the observed CpG spike at potential EGR binding sites in the no-peak group, which does not show signs of chromatin opening or TF binding (Figure 6A top left panel and examples in 6b) should also result from demethylation protection.

If TET2 would have access to the protected CpGs in the absence of EGR2 (perhaps, because it is recruited by a different factor), we would expect to observe the gradual demethylation of these sites (similar to the surrounding "unprotected" CpGs) over the seven days of culture. However, since CpGs overlapping the EGR2 binding site are almost completely protected from demethylation, the protective factor (EGR2) must be present, whenever TET2 has access to the DMR. This behavior is explained if EGR2 also recruits TET2 during transient binding.

20. Line 364-368. I do not follow this logic: "Hence, demethylation of CpGs around these binding sites only proceeds when the binding site is occupied." The CpG sites at these two different motif regions (EGR2 and AP-1) are presumably different regions of the genome. Where is the evidence that the binding site must be occupied? I get the second part of the sentence that the occupying factor needs to recruit demethylation machinery.

As already explained above, we argue that the protected CpGs within the EGR2 motif indicate EGR2 binding. Since the protected CpGs are not demethylated, the protecting factor needs to be present whenever TET2 is present. Since we show that EGR2 interacts with TET2 in CoIPs, EGR2 is the plausible factor responsible for recruiting TET2.

21. I also get lost in lines 405-417. In Figure 6A the authors make the claim that EGR2 blocks demethylation. Then in lines 405-417 they claim EGR2 does recruit TET2 to induce demethylation. Again I may have missed something.

As stated above, the block is restricted to the EGR binding motif sequences. When EGR2 binds to its consensus sequence, methylated CpGs are protected (or hidden) from the recruited TET2, which only gets efficient access to the surrounding sites. We have rephrased some of the corresponding sections in the main text to make this clearer.

22. Line 431 – please add reference.

We have added the reference.

23. Methods cell culture – while the authors reference their previous studies (2 in fact), it would still be useful for the reader to know briefly how monocytes were isolated: percol? Negative beads? Positive bead?

Blood MO were separated by leukapheresis of healthy donors followed by density gradient centrifugation over Ficoll/Hypaque and subsequent counter current centrifugal elutriation – no beads involved. We have now added this statement in the Methods section.

Reviewer #2 (Remarks to the Author):

This is a complex paper that has potential interest. The focus is on the mechanism underlying global DNA demethylation during monocyte differentiation. However, the definitive statement in the title: "The epigenetic pioneer EGR2 initiates DNA demethylation in differentiating monocytes at both stable and transient binding sites" is not proven by the data.

Specific comments

1. Gene names and protein names should be consistent with current nomenclature and format (CEBPA, KLF4, TCFP2L1, TET2 for protein)

We have corrected the protein names as suggested.

2. The rationale for the use of the GM-CSF/IL4 system is not provided. It is commonly recognized that monocyte-derived APC are distinct from classical DC; MDDC is the more common abbreviation. It is unclear whether IL4 is necessary or relevant in this system. GM-CSF stimulated monocytes are also considered as novel inflammatory macrophages (e.g. PMID:27525438) and CCL17 (which is one of the induced genes in Figure 2b) is a pro-inflammatory effector. The paper is really about monocyte differentiation and the end point is not that relevant. If iDC is used as an abbreviation, it should be better defined and justified with reference to the literature.

We are working with GM-CSF/IL4-induced MO-derived moDC for over 20 years. For us, it represents an ideal model, because these cells undergo marked transcriptional and epigenetic changes in the absence of cell division. In addition, the cells detach from the surface, which makes it a lot easier to handle them compared to extremely adherent macrophages. While part of the transcriptional program between GM-CSF and IL-4/GM-CSF may overlap, IL4 is actually required for the DC-like phenotype (see Sander et al. 2017, Immunity). These in vitro differentiated cells share some similarity with inflammatory DC (see Segura et al, 2013, Immunity).

But as this reviewer correctly points out, the endpoint of monocyte differentiation is not really relevant. We are interested in active demethylation turnover during MO differentiation and one could study this in any MO-derived population.

In response to both reviewers comments we re-written parts of the introduction and have now added some additional information regarding the culture system. We also changed the abbreviation iDC to moDC which is more commonly used in the field.

3. EGR2 is also induced by IFN γ in monocytes and is rapidly induced during monocytic differentiation of THP-1 cells in response to PMA. It is expressed by MDM as well as MDDC. So, it is not clear that it is specific to "DC-related transcriptional programs". As background to its proposed role as a pioneer it would be useful to know the time course of induction and nuclear accumulation in monocytes treated with GM-CSF/IL4.

Specificity was originally meant in the context of our differentiation model. Also, we have previously published the association of EGR2 with enhancers of MO-derived macrophages (Pham et al. 2012, Blood) and didn't actually claim that EGR2 would be specific to DC-related transcriptional programs. Specificity within all monocyte-derived cells (although interesting) has never been a topic and would be beyond the scope of this study.

To avoid misunderstandings, we have rephrased the text whenever the term "specific" may be misinterpreted.

We have also added new Western-blot images showing a time-course of EGR2 and IRF4 induction during GM-CSF/IL4-induced MO differentiation, which further supports their suggested role in moDCs. BTW.: In our hands (FANTOM consortium, Nat.Genetics 2009), THP-1 cells induce EGR1 and not EGR2 upon PMA-treatment.

4. It was not clear to me how “example regions” in Figure 1h and Supp Fig 1f were chosen and what they were meant to show. Needs justification and more description in the text. Perhaps worth noting from other data that CLEC10A and DNASE1L3 are specifically induced by GM-CSF but not by M-CSF. SLAMF1 and MAF on the other hand are not specific. The previous study (Ref 9) used CCL13 as an example; it is also strongly GM-CSF inducible.

The “example regions” show loci that are subject to DNA demethylation. We show DMR in genes that we had previously studied in earlier publications and two novel examples. In addition to the WGBS tracks (showing DNA methylation levels) we show 5hmC and ATAC tracks to correlate their changes over time at indicated DMR. In response to this reviewer’s comment, we added some highlighting and explanations in the main text and Figure legends to make this clearer.

5. Figure 1 is an analysis of methylation states from published data. The source publications and their context should be cited in the main text. It is misnomer to refer to DC-specific DMR. They are relatively demethylated only by comparison to monocytes. Many are likely also demethylated in response to M-CSF and/or in other cell types. It would actually be useful to identify the subset of DMR that are specific to MDDC versus MDM (grown in M-CSF).

In response to this comment we have added references for the published WGBS data (as well as sample numbers, as requested by reviewer 1). To accommodate both reviewers concerns, we have changed the references to “iDC-specific DMR” into “DMR demethylated in moDC” and “MO-specific DMR” into “DMR methylated in moDC”. As we pointed out above, the suggested type of would be interesting for moDC and MO-derived macrophages (not only for those generated with M-CSF). But this is clearly beyond the scope of our study.

6. “In line with the unidirectional DNA methylation turnover, regions that loose chromatin accessibility (as detected by ATAC sequencing) maintain their DNA methylation state while those that gain accessibility are demethylated (Supplementary Figure 1d)” (Spelling error and would be better in past tense).

We corrected the typo and changed to past tense.

7. Figure 2a indicates rather poor viability in the siRNA treatment and Figure S2b is too small and insufficient resolution to document unchanged morphology.

We are working with primary non-dividing cells and after 7 days of culture we always lose a certain fraction of cells. It may not have been clear that the % cell viability is given relative to the number of initially cultured monocytes – this is now explained in the Figure legend. To avoid the impression that the transfection impacts on viability, we now also included corresponding data from untransfected standard cultures in Figure 1A, which are not significantly different from the transfected ones. In response to this comment, we also generated new microscopy images with higher resolution and increased size in Supplementary Figure 2b. We hope that readers will now be able to judge the morphology of cells in culture.

8. Paragraph on bottom of page 5 has multiple syntax errors and needs to be re-written. the use of words like “few” and “mild” creates ambiguity; they mean different things to different people. I think the take home message is that the Tet2 knockdown selectively impacted only a small subset of the genes that were induced during differentiation. It is stated that “A large fraction of downregulated genes (22 in total) were normally induced during iDC differentiation (Figure 2b & S2d), indicating a mild but reproducible transcriptional effect of the TET2 knock-down”. Large and mild and reproducible are not defined. How reproducible was this fraction in multiple replicates? What is the likelihood of this occurring by chance based upon the number of genes that are induced during

monocyte differentiation (which is quite large). What proportion of genes that were induced were not impacted by TET2 knockdown.

In response to both reviewers comments, we have restructured and rephrased this paragraph. We moved the DNA methylation analysis in cells after TET2 knockdown to the top, since this generally confirms that the siRNA-mediated depletion of TET2 indeed delays DNA demethylation. The heatmap of gene expression data now includes all genes that are differential between knock-down and control cells. The independent MO/moDC data was removed and instead, the significant overlap with differentiation induced genes is now shown by GSEA (Figure 2D). As described in the methods section, differential gene expression analysis was done by linear modelling in edgeR and statistics are based on the analysis of all available replicates.

9. I may have missed the logic, but why is the set of 3 example genes in Figure 2d different from the set of 22 differentially regulated genes in Figure 2b.

We hope that the restructuring of this paragraph makes the logic clearer now. The regions shown are previously identified and established targets of demethylation and used as controls to confirm the effect of TET2 knockdown on demethylation. In Supplementary Figure 2f we focused on a few examples where accessibility is not or less established in TET2 knockdown cells. We did not focus on transcripts (as suggested by the reviewer) because it is not straightforward to link gene expression changes to regulatory elements (which are often distal).

10. p6 states “Gene expression data suggested that IFR4 (sic)and EGR2 are transcriptionally activated in iDC (Supplementary Figure 3b), qualifying them as candidate factors driving de novo demethylation processes.” It was not immediately clear from the legend or text where these data came from. In published data, IRF4 is induced in MDDC but not MDM; EGR2 is not specific to the GM-CSF treated cells.

The expression data is derived from RNA sequencing performed as part of this study. We have now added this information in the Figure legend. As stated previously, we never claim that EGR2 is specific for moDC relative to other types of MO-derived cells.

11. p6. Text states “Consistent with the slight loss of chromatin accessibility across DMR (Figure 2c), we observed a slight reduction of ATAC signals across PU.1 and AP1 motifs at DMR in TET2 knock-down samples (Supplementary Figure 3c)”. This does not seem compelling from the Figures. Can “slight” be quantitated and what is the FDR if one looks at a comparable size permuted set.

The footprints represent cumulative data and are difficult to “validate”. Since differences were subtle, we removed this Figure and corresponding text in the main manuscript.

12. The IRF4 knockdown experiment is not described sufficiently. It is not clear from the text or methods when the siRNA treatment was performed (on monocytes prior to differentiation?), how long afterwards viability was measured and by what method. Suppl Fig4B is not sufficient resolution to support the finding of “drastically altered cell morphology”. Suggest that since it is supplementary, larger images and a comparison with MDM grown in M-CSF would be more convincing. The data in Figure 4b/c is difficult to interpret and scaled data gives a misleading impression of magnitude. One might conclude from this figure that MDM lack class II MHC gene expression, which is certainly not the case in other datasets. The Z score is not defined. I could not find an accession number for the data. Supplementary Table 7 shows Acc XXX. For greater utility to allow others to replicate the analysis all the RNA-seq processed data (CPM or RKPM) should be available as Supplementary data in an Excel data sheet. The GO analysis gives little additional insight and could be removed. The set of genes apparently up-regulated (or not down-regulated) in the IRF4 knockdown is not discussed and they give clear insights

into possible mechanism, including transcription factors (PPARG, IRF8, CEBPB) and possible autocrine growth factor (CSF1).

All siRNA transfections were performed immediately after MO isolation prior to differentiation, which is now clearly stated in the Methods section. As requested, we have also added information on the viability testing which was routinely performed by trypan blue staining. We have now generated additional microscopy images with higher resolution, which hopefully clarify the major morphological differences, including the macrophage-like shape and adherence. In response to the first reviewer's comments, we have also redone Figure 4c (with corrected filtering and omission of gene labels) and provide additional GSEA to make our point clearer. We prefer to keep the pathway/Go analyses, because they highlight potentially functional differences between controls and knock-down cells.

All data sets have now been deposited (also see the first comment of reviewer 1) and the requested processed data underlying the presented Figures are available in the source data file.

As to the presentation of the gene expression data – scaling of the RNA seq data prior to the generation of heatmaps is a standard approach to focus on changes and not absolute values, which are difficult to present in a heatmap. As pointed out in the response to the first reviewer's point, we originally tried to avoid mixing data sets from independent experiments. However, to avoid the problem of independent scaling, we have now included all samples into the analyses. We also now include a set of bar charts showing rpkm expression values for selected genes that are significantly different between siCTRL and IRF4kd, allowing their direct comparison (new Supplementary Figure 4c).

In the last part of this comment the reviewer refers to a number of genes that should provide clear insights into possible mechanisms. We hope this reviewer agrees, that the obvious mechanism for the altered phenotype is the lack of IRF4. It is beyond the scope of this study to follow up on potentially relevant IRF4 target genes and since the observation period after knockdown is rather long (7 days), this reviewer might agree that it is impossible to distinguish expression changes that are a sole consequence of IRF4 knock-down or a consequence of secondary alterations in IRF4-target genes. Hence, we prefer not to discuss additional genes.

13. Same comments relate to the EGR2 knockdown but even more so. As presented, the effect is difficult to interpret. There is no obligate causal link between EGR2 knockdown and gene expression or methylation changes. How should we interpret an effect of the siRNA if 75% of cells are lost. Suppl Figure 5B is inadequate to describe the morphology of surviving cells. Are the EGR2 knockdown cells undergoing apoptosis? Is this simply an analysis of the selected subset of cells (monocytes or MDDC?) that survive the EGR2 knockdown and perhaps did not express EGR2 anyway? Is the expression of EGR2 heterogeneous in unselected population. This interpretation is supported by the observation that “Genes upregulated in MO cultures upon EGR2 knock-down also included a number of T cell specific marker genes, suggesting that contaminating cells were unaffected (and hence enriched) by the EGR2 knock-down”. The set of genes that is apparently up-regulated by the EGR2 knockdown looks like a monocyte signature (IRF8, CEBPB, S100A8/9, CD14) and again is not discussed. It may be that undifferentiated monocytes are the only cells that survive. It may still be possible to infer a relationship between EGR2 binding and demethylation but with significant caveats.

We agree with this reviewer that the phenotype of EGR2 knockdowns are more difficult to interpret, but we disagree that there is “no obligate causal link between EGR2 knockdown and gene expression or methylation changes”, as stated in the above comment.

We lose roughly 50% of the cells that normally survive the culture and because contaminating T cells are not affected by the knockdown, we observe upregulation of T cell signature genes, which is clearly stated. This is a general problem arising from such phenotypes in primary cell cultures. We have revised the text to make this clearer and now indicate that the interpretation of the expression data may be limited by the strong phenotype. In addition, we

added bar plots in Supplementary Figure 5e presenting the gene expression data (rpkm values) of exemplary myeloid and lymphoid TF in control and knockdown cells, suggesting that the expression profiles are still dominated by MO-derived cells.

We also provide novel microscopy images at higher magnification showing that while some of the surviving cells develop the typical DC-like shape, the observed debris suggest that the other cells indeed undergo apoptosis. Hence, based on their morphology, there is no evidence that the remaining cells would be “arrested” in a monocyte-like state, as suggested by the reviewer. In fact, there is no published evidence and we have never observed “undifferentiated” monocytes after 7 days in any type of culture system before. In addition, we have added IGV genome browser tracks of ATAC data across genes that are highly induced during moDC differentiation (new Supplementary Figure 5f). Here, we clearly see that moDC typical changes in chromatin accessibility also occur in the absence of EGR2.

As an alternative interpretation of the knock-down data, the reviewer suggests that our culture may be heterogenous regarding EGR2 expression and that the remaining cells after culture derive from monocytes that do not induce EGR2. While we have not tested this and cannot rule out this possibility, this would not change any conclusions regarding the interpretation of chromatin accessibility or methylation data, which is that EGR2 is required to open up a subset of regions (which are also characterized by a strongly enriched EGR consensus motif) and to demethylate these sites. If EGR2 is absent (and this can either be because we knocked it down or because the cells never expressed it, as suggested by this reviewer), the sites are never opened or demethylated. Hence EGR2 is required no matter why it is absent.

For reasons stated in the previous comment and given the difficult interpretation of gene expression results (as pointed out by this reviewer), we do not feel that we can meaningfully discuss individual differentially expressed genes as suggested by this reviewer. However, the ATAC data clearly links chromatin accessibility-losses with EGR2 motifs, suggesting that we specifically lose EGR2 binding sites in EGR2 knock-down cells. The same sites also clearly maintain their methylation state, hence, EGR2 is clearly required for active DNA demethylation.

14. p11. “The almost complete resistance to demethylation suggested that the demethylation machinery does not have access to the protected CpGs. Hence, demethylation of CpGs around these binding sites only proceeds when the binding site is occupied, and the occupying factor must be critical for recruiting the demethylation machinery. “ The logic is not clear to me and the Figure 6A is at insufficient size, resolution and color contrast to decipher the intended message.

In response to both reviewers comments, we altered the coloring (to better distinguish MO and moDC data sets) and added a smaller inlay figure zooming into the graph and showing DNA methylation values across the transcription factor motif sequences only. We hope that the lack of demethylation at the indicated CpGs within the EGR2 motif is now obvious. As stated in the main text, the “demethylation protection” is restricted to the CpGs overlapping the 10-12 bp binding site, which may have not been clearly visible in the original images. As outlined in our response to reviewer 1 comments 19-21, we have also changed the text to make the logic clearer.

15. The discussion provides an unconvincing explanation for the lack of effect of the TET2 knockdown on demethylation. Their previous study concluded that TET2 was the major demethylase in monocytes and TET3 was less highly-expressed. There is a recent literature on TET3 in monocytes and it is certainly detectable at the mRNA level in published data. Could redundancy be an alternative explanation? The authors might comment on detection of TET3 in their own RNA-seq data.

It is not entirely clear to us, why this reviewer thinks that our original explanation is unconvincing. In our setting of primary, non-dividing cells, the effect of any siRNA will be delayed depending on protein stability. Many demethylation events are initiated early, upon culture of the cells, when TET2 protein levels are still high even after siRNA-mediated knock-down. Hence, we did not expect that our approach would completely prevent active DNA

demethylation. In fact, we usually see a reduction of demethylation by about 50% in siRNA treated cells (s Figure 2b and Supplementary Figures 2c). In the reverse argument, most DMR are still demethylated in 50% of the knock-out cells. Hence the expected effect on transcriptional level is ~50%, provided that the lack of demethylation completely blocks transcription. Since most DMR represent distal regulatory elements, which often have more subtle effects on gene expression, the effect on transcription may even be smaller and hence difficult to measure.

Nevertheless, the comment on TET3 is valid and we now provide the corresponding gene expression data in Supplementary Figure 2f. TET2 levels in MO are much higher on mRNA level compared to TET3, but mRNA expression levels of TET2 decline over time, which may point to a role for TET3 in compensating for TET2 depletion during MO differentiation. As suggested by this reviewer, we have now included this possibility in our discussion.

16. The discussion finishes on a bit of an afterthought.

In response to this comment, we have added a concluding sentence at the end of our discussion.

Reviewer #1 (Remarks to the Author):

The authors adequately addressed my comments.

Reviewer #2 (Remarks to the Author):

The authors have provided a comprehensive and compelling response to previous review and the manuscript is greatly improved.

There is a minor spelling error in line 350 (loose should be lose).

It is a semantic choice, but I would suggest that parts of the abstract reporting the actual results would be better in past tense.

Response to the final comments of reviewers:

Point-by-point response:

Reviewer #2 (Remarks to the Author):

The authors have provided a comprehensive and compelling response to previous review and the manuscript is greatly improved.

There is a minor spelling error in line 350 (loose should be lose).

The typo has now been corrected

It is a semantic choice, but I would suggest that parts of the abstract reporting the actual results would be better in past tense.

As the reviewer points out, this is a semantic choice – if the editor feels that we should change the abstract, we will be happy to change it. Otherwise we would like to leave it as it is....